# Persistent neural activity in auditory cortex is related to auditory working memory in humans and nonhuman primates

Ying Huang[1]*, Artur Matysiak[2], Peter Heil[3,4], Reinhard König[2], Michael Brosch[1,4]

[1]Special Lab Primate Neurobiology, Leibniz Institute for Neurobiology, Magdeburg, Germany; [2]Special Lab Non-Invasive Brain Imaging, Leibniz Institute for Neurobiology, Magdeburg, Germany; [3]Department Systems Physiology of Learning, Leibniz Institute for Neurobiology, Magdeburg, Germany; [4]Center for Behavioral Brain Sciences, Otto-von-Guericke-University, Magdeburg, Germany

**Abstract** Working memory is the cognitive capacity of short-term storage of information for goal-directed behaviors. Where and how this capacity is implemented in the brain are unresolved questions. We show that auditory cortex stores information by persistent changes of neural activity. We separated activity related to working memory from activity related to other mental processes by having humans and monkeys perform different tasks with varying working memory demands on the same sound sequences. Working memory was reflected in the spiking activity of individual neurons in auditory cortex and in the activity of neuronal populations, that is, in local field potentials and magnetic fields. Our results provide direct support for the idea that temporary storage of information recruits the same brain areas that also process the information. Because similar activity was observed in the two species, the cellular bases of some auditory working memory processes in humans can be studied in monkeys.

*For correspondence: Ying.
Huang@ifn-magdeburg.de

**Competing interests:** The authors declare that no competing interests exist.

## Introduction

Working memory (WM) has been defined as a cognitive process for temporary storage of task-relevant information for goal-directed behaviors (*D'Esposito, 2007*; *Sreenivasan et al., 2014*). The information can be related to past sensory events (e.g., short-term sensory memory), or to future sensory events (e.g., sensory prediction) or actions. The two types of information have been termed the retrospective and prospective codes, respectively, and both can be stored in WM to bridge sensory events or their contingent behavioral actions (*Curtis et al., 2004*; *D'Esposito, 2007*; *Postle, 2006*; *Sreenivasan et al., 2014*). Research on WM has promoted the view that the storage is accomplished by sustained attention to internal representations of information (*Awh and Jonides, 2001*; *Chun, 2011*; *Gazzaley and Nobre, 2011*; *Kiyonaga and Egner, 2013*; *Postle, 2006*; *Zimmermann et al., 2016*). It is further assumed that WM arises through the coordinated recruitment, via attention, of brain areas in a broad network (*Constantinidis and Procyk, 2004*; *Postle, 2006*; *Ranganath and D'Esposito, 2005*) and that the task-relevant information is stored in some of the same brain areas that also process it, such as sensory cortex, whereas prefrontal cortex (PFC) is thought to aid storage of the information in sensory cortex (*D'Esposito, 2007*; *Postle, 2006*; *Sreenivasan et al., 2014*).

In the auditory modality, lesion (*Colombo et al., 1990*, *1996*; *Fritz et al., 2005*), imaging (*Brechmann et al., 2007*; *Grimault et al., 2014*; *Guimond et al., 2011*; *Kumar et al., 2016*;

*Linke and Cusack, 2015*; *Linke et al., 2011*; *Nolden et al., 2013*; *Rämä et al., 2004*), and recording studies (*Bigelow et al., 2014*; *Gottlieb et al., 1989*; *Sakurai, 1994*; *Scott et al., 2014*) have revealed that auditory cortex (AC) is involved in the performance of auditory WM tasks. Although some of these studies have demonstrated neural activity in AC that is persistently elevated or suppressed during the period when information needs to be held in WM, we argue here that such persistent changes in activity do not unequivocally reflect WM. Many studies have not controlled for potential long-lasting neural activity evoked by a stimulus and for mental processes other than WM that are associated with performing a task, such as general attention, reward expectation or preparation for behavioral responses. These processes can also be associated with changes in AC activity that last for seconds (*Brosch et al., 2011*). For example, in some of the imaging (*Kumar et al., 2016*; *Linke and Cusack, 2015*) and recording studies (*Bigelow et al., 2014*; *Gottlieb et al., 1989*; *Scott et al., 2014*), persistent changes in activity were revealed by comparing activity during the WM period with that during a baseline period. However, these two periods differed not only in WM but also with respect to the expectation of upcoming stimuli and of rewards, and with respect to preparation for behavioral responses. Therefore, the persistent changes in activity revealed in these studies do not necessarily reflect WM. They could reflect expectation and preparation. In some other studies (*Grimault et al., 2014*; *Nolden et al., 2013*), the persistent changes in activity were revealed by comparing activity in experimental conditions with different WM load. However, in these studies, the auditory stimuli always co-varied with the WM load across conditions. Therefore, differences in the persistent activity revealed in these studies could reflect differences in activity evoked by different stimuli rather than differences in WM load. Moreover, it is unclear whether persistent changes in neural activity in AC are stimulus specific (*Gottlieb et al., 1989*; *Kumar et al., 2016*; *Lemus et al., 2009*; *Linke and Cusack, 2015*; *Linke et al., 2011*; *Rämä et al., 2004*; *Scott et al., 2014*), even though stimulus specificity has been traditionally considered a hallmark of WM in sensory cortex (*Curtis and Lee, 2010*). Furthermore, with few recent exceptions (*Grimault et al., 2014*; *Nolden et al., 2013*), human studies have not attempted to localize neural activity related to the performance of auditory WM tasks (e.g., *Guimond et al., 2011*; *Kaiser et al., 2009*; *Lu et al., 1992*). Thus, it is still an open question whether AC exhibits persistent changes in neural activity related to auditory WM.

To address this issue, we conducted three studies on humans and nonhuman primates in which they performed different tasks on sequences of two sounds, S1 and S2, separated by a delay (*Figure 1*). The tasks and the sequences differed in the extent to which WM was required. They were designed to separate neural activity in AC related to WM from potential late activity evoked by S1 or activity related to mental processes other than WM. They also allowed examination of the stimulus specificity of WM-related activity in AC. In humans, we assessed neural activity in AC by means of magnetoencephalography (MEG), through which we determined the strengths of regional sources (*Scherg and Ebersole, 1993*). In monkeys, we recorded local field potentials (LFPs) and spike activity from core fields of AC. To track potential contributions from other brain structures, particularly from PFC, we not only compared LFPs and spike activity in core fields of AC but also recorded spike activity in PFC.

## Results

### Study 1: Working-memory related activity in human auditory cortex revealed by contrasting auditory working memory tasks with a delayed-response task

#### Experimental overview and rationale

Human subjects performed three tasks on an identical set of four sequences of tones. The tasks and the sequences differed in the extent to which WM was required. Each sequence was composed of two tones, S1 and S2, separated by a delay (*Figure 1a,c*). The tones could be of frequency A or B, resulting in the sequences AA, AB, BA, and BB.

Tasks 1 and 2 both required a go response (a button press within a limited time window after S2) to the sequence AA and a nogo response (no button press) to the other three sequences (*Figure 1a*). The two tasks differed with respect to the index finger used for the go response: the right for task 1 and the left for task 2. They also differed with respect to the ear stimulated: the left

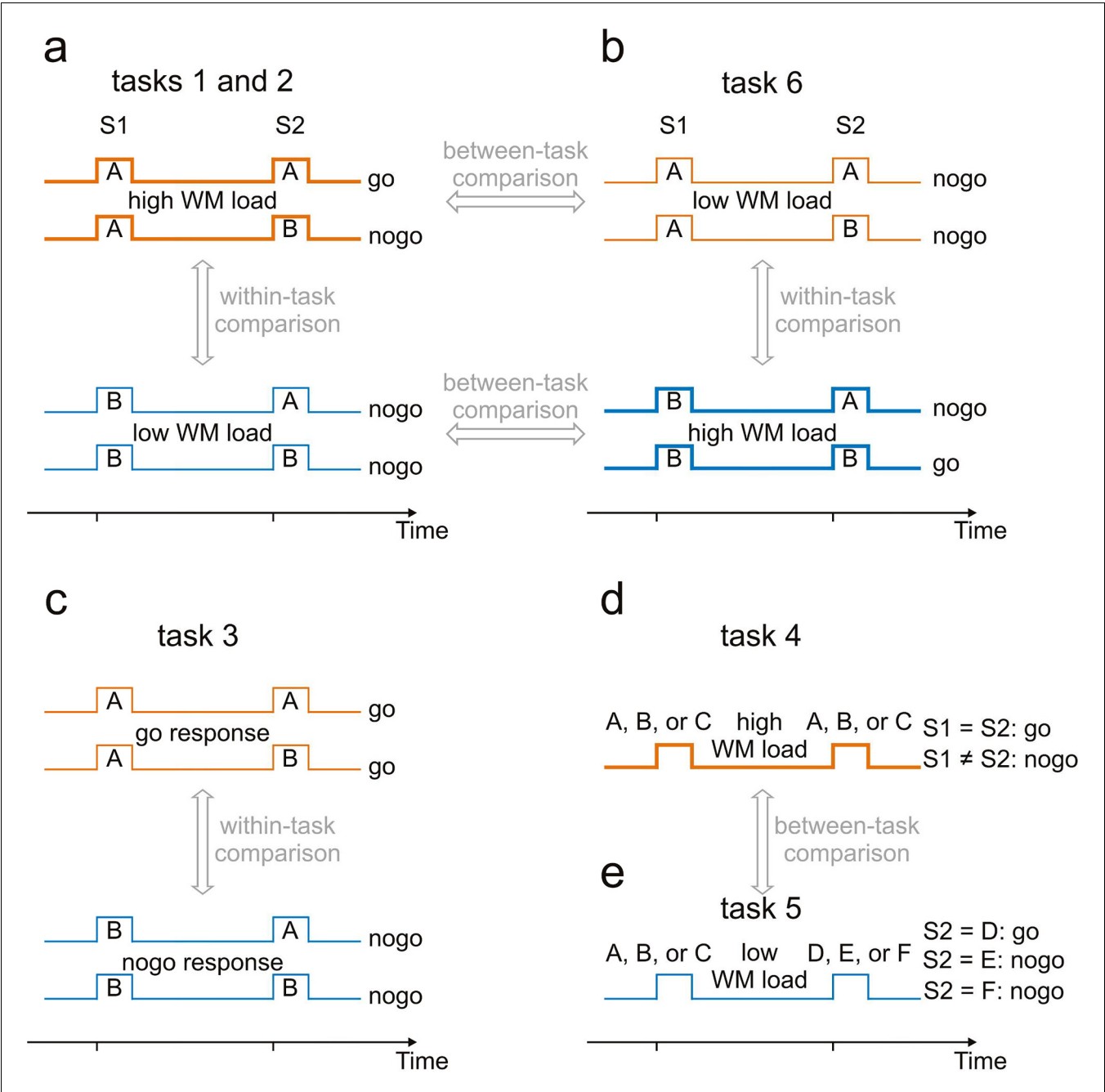

**Figure 1.** Experimental paradigms used to identify working-memory related neural activity. A sequence of two stimuli, S1 and S2, separated by a delay, was presented on every trial. Tasks associated with these sequences differed. For tasks 1, 2, 3 and 6, four sequences (AA, AB, BA and BB) were used. A and B represent tones of different frequencies. For task 4, S1 and S2 were tones of three different frequencies, A, B or C. For task 5, S1 was also a tone (of frequency A, B or C), whereas S2 was a noise burst (D), a frequency-modulated tone (E) or a click train (F). (**a**) Tasks 1 and 2 required a go response for sequence AA and a nogo response for the other sequences. The WM load was higher when S1 was A (thick orange traces) than when it was B (thin blue traces). Tasks 1 and 2 differed with respect to the hand used for the responses. Putative WM-related activity was identified by a within-task comparison. (**b**) Task 6 required a go response for sequence BB and a nogo response for the other sequences. The partial changes of stimulus-response associations relative to those of task 1 enabled WM-related activity to be revealed also by between-task comparisons. (**c**) Task 3, a delayed-response task, required a go response when S1 was A and a nogo response when it was B, irrespective of the frequency of S2. Potential differences in activity related to preparation for go and nogo responses could be revealed by comparing activity during the delay of trials requiring go responses (thin orange traces) and nogo responses (thin blue traces). (**d**) Task 4, a delayed-match-to-sample task, required a go response when S1 and S2 were identical and a nogo response otherwise. (**e**) Task 5, a sound-discrimination task, required a go response when S2 was the noise burst and a nogo response when S2 was the click train or the frequency-modulated tone, irrespective of the frequency of S1. WM-related activity was revealed by

*Figure 1 continued on next page*

*Figure 1 continued*

comparing activity during the delay in tasks 4 and 5. Feedback of whether the response was correct or not was provided to the human subjects in tasks 4 and 5 and to the monkeys in tasks 1 and 6. The human subjects were shown a smiling face immediately after a correct response and a frowning face after an incorrect response. For the monkeys, a drop of water was provided immediately after correct go and nogo responses.

The following figure supplements are available for figure 1:

**Figure supplement 1.** Schemes of source positions used for the analysis of the MEG data.

**Figure supplement 2.** Behavioral performance of monkeys C and L in the working-memory tasks.

**Figure supplement 3.** Demonstration of the recording area in prefrontal cortex.

for task 1 and the right for task 2, contralateral to the finger used. The tasks thus controlled for potential differential effects of the side of auditory stimulation and execution of the behavioral response and allowed the investigation of whether both the left and right AC are involved in auditory WM. In order to solve tasks 1 and 2, subjects first had to identify the frequency of S1 because it determined how the remainder of the trial needed to be processed. When S1 was B, a nogo response was required, irrespective of S2, and nothing else needed to be held in WM. When S1 was A, however, subjects remained uncertain whether a go or a nogo response would be required after S2. To cope with this uncertainty, they could use different strategies involving WM. For example, they could hold in WM the two possible associations between S2 and the behavioral response (a go response when S2 was A and a nogo response when it was B), and then make the appropriate response after identifying S2. Or they could hold in WM the frequency of S1, compare it with that of S2, and then make a go or a nogo response depending on whether S2 was identical to or different from S1, respectively. Irrespective of the strategies subjects utilized, WM was involved during the delay between S1 and S2 in each trial. The random presentation of the four sequences, AA, AB, BA and BB, prevented the predictability of the sequence in the upcoming trial and, consequently, the information required for making a correct response needed to be stored and updated on a trial-by-trial basis, that is, in WM on a scale of seconds (*Goldman-Rakic, 1995*). Because more information needed to be stored during the delay in trials when S1 was A than when it was B, the WM load during delay was higher in the former than in the latter trials. In the following, these trials are referred to as high- and low-WM-load trials, respectively.

If AC were involved in WM, there should be differences in AC activity during the delay between high- and low-WM-load trials, but the presence of such differences would not allow the conclusion that they are due to differences in WM load. They could also result from differences in WM-unrelated activity that is evoked by S1 but outlasts it (*late activity*) or from differences related to preparing for a possibly required go response within a limited time window after S2 in high-WM-load trials rather than for the nogo response in low-WM-load trials (*preparatory activity*). To distinguish between these possibilities, the same subjects also performed task 3 on the same stimuli and used the same finger as in task 1 (*Figure 1c*). Task 3 was a delayed-response task in which subjects had to hold in WM that, irrespective of the identity of S2, a go response after S2 was required when S1 was A and a nogo response was required when S1 was B. If differences in activity during the delay in task 1 were only due to differences in late activity or in preparatory activity, then the differences in activity between the high- and low-WM-load trials in task 1 should be very similar to the differences between the go and nogo trials in task 3. If, however, such a similarity is not observed, then the differences in activity during the delay in task 1 (and by inference task 2) are at least partly due to differences in WM load.

Subjects performed tasks 1, 2 and 3 in three separate blocks within one experimental session of MEG measurements. Each block consisted of 240 trials (60 trials for each sequence) and lasted ~20 min. The order of the blocks was randomized across subjects. The frequencies of tones A and B were 1.5 and 1.6 kHz, respectively. The tone duration was 100 ms and the stimulus-onset interval 2000 ms. Neural activity in AC was estimated by determining the strengths of two regional sources. They were seeded in the left and right AC, respectively, based on the anatomical MR image of each subject's brain. To obtain the grand mean source waveforms, individual source waveforms were

geometrically averaged across subjects, for reasons given elsewhere (*König et al., 2015*; *Matysiak et al., 2013*; *Zacharias et al., 2011*). Source strengths were analyzed for correct trials (~90%) only.

## Results of study 1

We found neural activity in the human AC that was related to WM. *Figure 2a and b* show that in task 1, the strength of the regional sources seeded in the left and right AC increased in response to S1 and, throughout the entire delay, remained at a level higher than that before S1 (p<0.001, permutation test). In addition, sources in the left and right AC were significantly stronger in high- (thick orange traces) than in low-WM-load trials (thin blue traces) during the final 500 ms of the delay (p<0.05 with Bonferroni correction for multiple comparisons, permutation test; for p-values, see the numbers above the abscissae in *Figure 2a,b*). In contrast, the peak source strengths around the time of the M100, the most prominent component of the early responses evoked by the two different S1s, were very similar in the high- and low-WM-load trials. Similar results were obtained in task 2 (*Figure 2c,d*) where the sounds were presented to the opposite ear and subjects used the opposite index finger for the go responses, although the differences between the high- and low-WM-load trials during the final 500 ms of the delay were only significant at the level of 0.05 without Bonferroni correction.

It is likely that the differences in source strength between the high- and low-WM-load trials in tasks 1 and 2 were due to differences in WM load but presumably not due to differences in preparatory activity or late activity. This conclusion is supported by the results of task 3 (*Figure 2e,f*). In this task, differences between go and nogo trials were not significant (p>0.05, permutation test), neither in the left AC (*Figure 2e*) nor in the right AC (*Figure 2f*).

## Study 2: Working-memory related activity in human auditory cortex revealed by contrasting a delayed-match-to-sample task with a sound-discrimination task

### Experimental overview and rationale

Study 2 aimed at finding independent support for the existence of WM-related activity in the human AC with new subjects and with an alternative approach that allowed the direct demonstration of WM-related activity by comparing trials that differed only in WM load and not in other characteristics including frequencies of S1 and preparation for motor responses. We contrasted a conventional delayed-match-to-sample task (task 4) with a sound-discrimination task (task 5). In task 4 (*Figure 1d*), tones of three different frequencies (A, B and C) were used as S1 and S2. A go response was required when the frequencies of S1 and S2 were identical and a nogo response otherwise. Subjects could solve this task by using similar strategies as in tasks 1 and 2 of study 1 for trials starting with A. All trials in task 4 constituted high-WM-load trials and a third of them required go responses. Task 5 (*Figure 1e*) required a go response when S2 had a specific identity and a nogo response otherwise, irrespective of the identity of S1. This specific S2 occurred on one third of the trials so that the proportions of go and nogo trials and, thus, the proportion of trials on which subject prepared for one or the other motor responses, were the same as in task 4. The WM load during the delay between S1 and S2 in task 5 was lower than that in task 4 and therefore all trials in task 5 were considered to be low-WM-load trials. For S1 in task 5, the same tones A, B and C were used as in task 4. Unlike in task 4, acoustic stimuli different from pure tones were used as S2: a noise burst (D), a click train (E), and a frequency-modulated tone (F). Subjects were instructed to make a go response when S2 was a noise burst (AD, BD and CD).

In both tasks, the frequencies of A, B and C were set to 1.5, 0.789 and 2.85 kHz, respectively. Subjects performed tasks 4 and 5 in separate blocks within one experimental session. Each block consisted of 360 trials (40 trials for each of the nine sequences) and lasted ~30 min. The order of the blocks was randomized across subjects. Source strengths were computed for correct trials (~95%) only.

### Results of study 2

We found WM-related activity in both the left and right AC by comparing the source strengths during the delay in tasks 4 and 5. *Figure 3* shows, for each hemisphere, the grand mean source

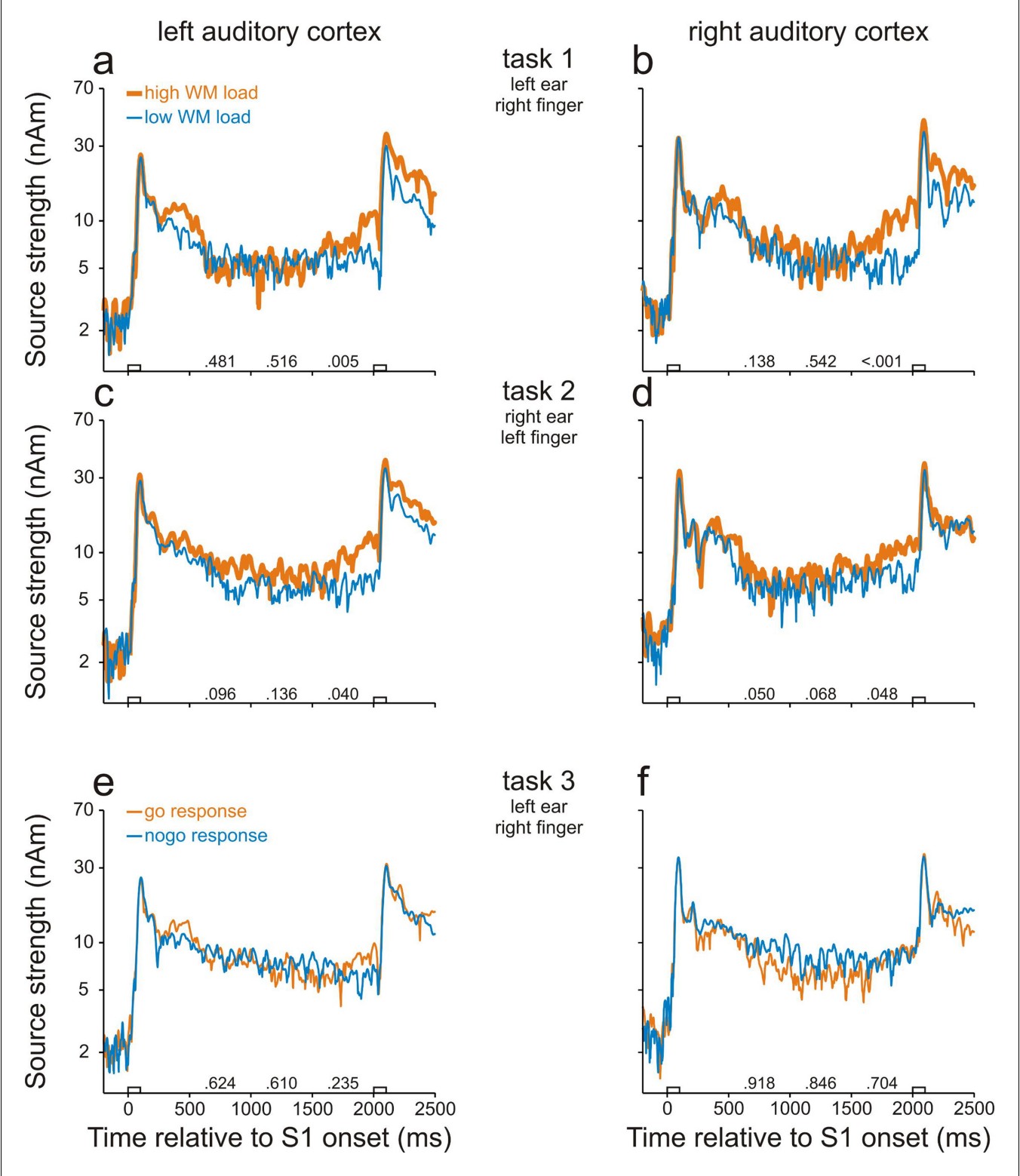

**Figure 2.** Working-memory related neural activity in human auditory cortex identified in study 1. Strengths of the sources in AC in task 1 (**a**, **b**), task 2 (**c**, **d**) and task 3 (**e**, **f**). Data from the left and right hemisphere are shown in the left and right column, respectively. Each trace represents the grand geometric mean source waveform derived from MEG recordings in 12 subjects. Thick orange and thin blue traces in **a–d** represent source strengths in high- and low-WM-load trials, respectively. Thin orange and thin blue traces in **e–f** represent source strengths in go and nogo trials, respectively. The empty bars on the abscissae represent the timing and duration of S1 and S2. The numbers on the abscissae are the p-values of permutation tests for

*Figure 2 continued on next page*

*Figure 2 continued*

the ratios of the source strength in high- to that in low-WM-load trials or in go to that in nogo trials during the three 500-ms periods of the delay. Note that the ordinates have logarithmic scaling. Also note that the differences in the left and right AC in tasks 1 and 2 reflect differences in WM load.

waveforms from all trials in task 4 (thick orange traces) and task 5 (thin blue traces). As in study 1, activity in AC was stronger than before S1 throughout the entire delay (p<0.001, permutation test) in both the delayed-match-to-sample (task 4) and the sound-discrimination task (task 5). In addition, in both ACs, the sources were stronger in task 4 than in task 5 during the entire delay, being significant during many portions of the delay (p<0.05 with Bonferroni correction, permutation test; for p values, see the numbers above the abscissae). In contrast, the peak source strengths during the M100 component evoked by S1 were very similar in the two tasks. This suggests that levels of general attention in tasks 4 and 5 were similar during the presentation of S1 because the peak amplitude of the M100 component is strongly modulated by attention (*Kauramäki et al., 2012*; *Woldorff et al., 1993*). Because tasks 4 and 5 differed in WM load, but not in S1 or in the proportions of go and nogo trials, we conclude that the differences in source strengths during the delay between the two tasks were due to differences in WM load and that activity in both ACs can be related to WM.

## Study 3: Working-memory related activity in core fields of monkey auditory cortex

### Experimental overview and rationale

In order to search for WM-related activity in core fields of AC at the level of single neurons and small groups of neurons and also in a different species, two monkeys (L and C) were each trained to perform two different tasks (tasks 1 and 6) on the same set of stimuli. Task 1 was equivalent to task 1 performed by the human subjects. It required a go response (a bar release) for the sequence AA and a nogo response (withholding a bar release) for the other three sequences AB, BA and BB (*Figure 1a*). Our analyses of the behavioral data (*Figure 1—figure supplement 2*) indicated that the monkeys solved the task utilizing WM and not solely by means of other strategies. Therefore, the

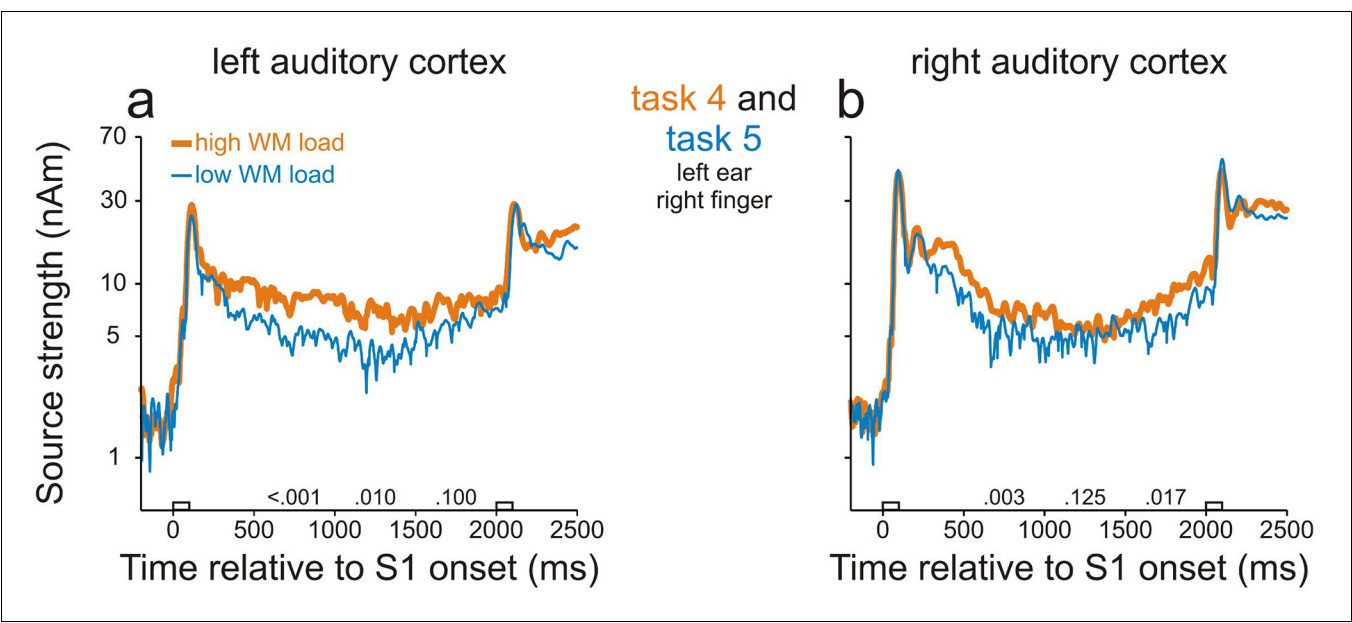

**Figure 3.** Working-memory related neural activity in human auditory cortex identified in study 2. Sources in the left (**a**) and right (**b**) AC were stronger in task 4 in which WM load was high (thick orange traces) than in task 5 in which WM load was low (thin blue traces). Each trace represents the grand geometric mean source waveform across 15 subjects. Other conventions as in *Figure 2*.

WM load during the delay was higher in trials when S1 was A than when it was B (for detailed reasoning, see the second paragraph in Section Experimental overview and rationale).

To rule out differences in late activity evoked by S1, or preparatory activity for the behavioral responses as potential confounds (see Section Experimental overview and rationale), the two monkeys also performed task 6, which was equivalent to task 1 except that the WM loads of the A and B tones were reversed. Task 6 required a go response for the sequence BB and a nogo response for the other three sequences (*Figure 1b*). We contrasted task 1 with task 6, rather than with a delayed-response task (task 3), because the reversal of WM load also allowed addressing the stimulus-specificity of WM-related activity. To further examine potential sources of differences in activity during the delay of high- and low-WM-load trials, the same set of stimuli was also presented to the two monkeys while they did not perform the tasks (*passive trained condition*). The stimuli were also presented to a third, untrained, monkey (*passive untrained condition*).

As in study 1, we identified WM-related activity by comparing activity in trials from the same task that started with A versus those that started with B (*within-task comparison*). If, at a given recording site in core fields of AC, differences in activity during the delay between these trials were only due to late activity that was evoked by S1 but outlasted it, one would expect very similar differences in task 1, in task 6, and in the passive conditions. If the differences were only due to preparatory activity, one would expect opposite differences in tasks 1 and 6 and no differences in the passive conditions. Therefore, if the differences were observed in only one task but not in the other, they would be due to differences in WM load and thus indicate WM-related activity. As a second method of identifying WM-related activity, we compared trials starting with the same tone but taken from different tasks (*between-task comparison*). Results of this method (*Figure 4—source data 1*) were consistent with those of the first method. As a third method of identifying WM-related activity, we compared high- and low-WM-load trials with regard to activity differences between the delay and the baseline within the same trial. In addition to WM-related activity, this method also revealed how the neuronal activity varied in an ongoing trial with WM load.

A and B were tones of 3 kHz and 1 kHz, respectively, with a duration of 200 ms. In most experimental sessions, the stimulus-onset interval between S1 and S2 was 1000 ms, resulting in an 800-ms delay, defined as the interval from the offset of S1 to the onset of S2. In other sessions, it was 1300 ms, resulting in an 1100-ms delay. Within each session a monkey performed tasks 1 and 6 in separate, alternating blocks (2–8 per session). A block consisted of ~140 trials and lasted ~25 min. Microelectrode recordings of single-unit and multiunit activity and of local field potentials from core fields of the right AC and, later in separate experimental sessions, also from the left prefrontal cortex started once the monkeys responded correctly to each sequence in ≥65% of the trials in both tasks during 10 consecutive sessions.

## Spike activity in core fields of auditory cortex related to working memory

The spike rates of many neurons in core fields of AC during the delay differed between high- and low-WM-load trials while the monkeys performed task 1 or task 6. *Figure 4a and b* show the average spike rates of two representative multiunits recorded from monkeys C and L while they performed task 6 correctly. In both examples, the differences in spike rate emerged shortly after the offset of S1 and persisted until the end of the delay. We compared the responses by computing the ratio of the spike rate in high- to that in low-WM-load trials (*high-/low-load spike-rate ratio*). Comparisons of the responses during the delay were limited to the final 500 ms to minimize the potential difference in late activity that was evoked by the S1s. Permutation tests revealed that the spike-rate ratios during the delay differed significantly from 1 ($p < 0.05$; green bars on the abscissae in *Figure 4a,b*). This was not the case during the 500-ms period directly before S1 (*baseline*). The spike rate during the delay was higher in the high- than in the low-WM-load trials in the example of *Figure 4a* and lower in the example of *Figure 4b*. In the following, we argue that such differences were due to differences in WM load.

*Figure 4c* shows the high-/low-load spike-rate ratios for all multiunits (124 in monkey L and 183 in monkey C) from the trials in which the two monkeys performed the two tasks correctly. The ratios ranged from 0.75 to 1.9. Ratios significantly different from 1 (*significant ratios*) were obtained from 27.4% and 34.5% of the multiunits during the delay in tasks 1 and 6, respectively (green dots), with the majority of ratios >1 (71 of 84 in task 1; 99 of 106 in task 6). These proportions are higher than

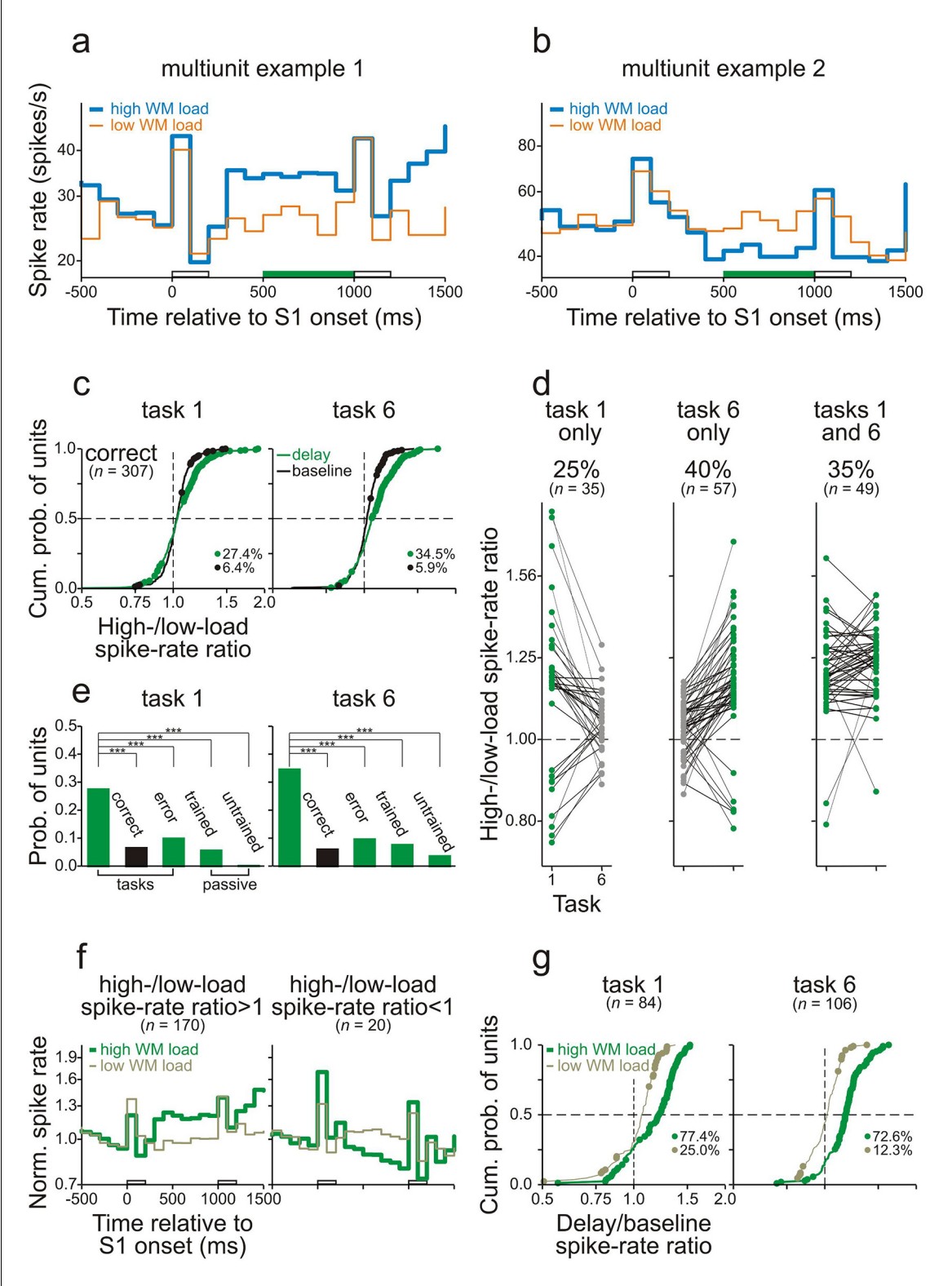

**Figure 4.** Working-memory related spike activity in monkey auditory cortex. (**a, b**) The spike rates of two representative multiunits recorded in AC while the monkeys performed task 6 correctly. Thick and thin traces represent spike rates in high- and low-WM-load trials, respectively. S1 was 1 kHz (blue) in high-WM-load trials and 3 kHz (orange) in low-WM-load trials. The green bars on the abscissae represent the final 500 ms of the delay during which the high-/low-load spike-rate ratios differed significantly from 1. In the example in panel a, the spike rate is higher, and in the example in panel b lower, in high- than in low-WM-load trials. The empty bars on the abscissae represent the timing and duration of S1 and S2. (**c**) The high-/low-load spike-rate

*Figure 4 continued on next page*

*Figure 4 continued*

ratios during the final 500 ms of the delay (green) and during the 500 ms directly before S1 (*baseline*; black) for all multiunits. Significant ratios are marked by dots. The proportions of units with significant ratios are also provided. Left and right panels show the ratios obtained when the monkeys performed task 1 and task 6 correctly. The horizontal dashed lines mark the median cumulative probability and the vertical dashed lines the ratio of 1. (d) Task and stimulus specificity of high-/low-load spike-rate ratios during the final 500 ms of the delay. Ratios obtained from a given unit in tasks 1 and 6 are connected by a line. Spike-rate ratios which are significantly different from 1 are represented by green dots, those which are not by gray dots. The left, middle, and right panels show the data from all units where the ratios were significantly different from 1 in task 1 only, in task 6 only, or in tasks 1 and 6, respectively. The 65% of units with significant ratios in one task only are considered to be involved in WM. The proportions and the numbers of units in the three groups are also provided. (e) Dependence of the occurrence of significant high-/low-load spike-rate ratios on the behavioral context. The green bars show the probability of units with significant ratios during the delay of correct trials, of error trials, and during passive conditions in the two trained monkeys and in the untrained monkey. The black bar shows the probability of units with significant ratios during the baseline of correct trials. The asterisks indicate significant differences between the conditions (chi-square test, one-tailed, p<0.001). (f) The mean spike rates in high- (thick traces) and low-WM-load trials (thin traces) for units with high-/low-load spike-rate ratios significantly >1 (left panel) or <1 (right panel). They were obtained by geometric averaging after normalizing each response to baseline. (g) Dominance of delay enhancement in AC. Distributions of the delay/baseline spike-rate ratios are shown separately for high- and low-WM-load trials and for task 1 and task 6. Ratios >1 represent delay enhancement and ratios <1 delay suppression. Significant ratios are marked by dots. Analyses were performed only on units with significant high-/low-load spike-rate ratios during the final 500 ms of the delay (panel c). The proportions of units with significant delay/baseline spike-rate ratios are also provided for high- and low-WM-load trials, respectively.

The following source data and figure supplement are available for figure 4:

**Source data 1.** Working-memory related neuronal activity identified by between-task comparison.

**Figure supplement 1.** Working-memory related spike activity of single units in monkey auditory cortex.

those expected by chance with a criterion of p<0.05 used in the permutation test and higher than the proportions of significant high-/low-load spike-rate ratios during the baseline (black dots in *Figure 4c*; ~6.0%; p<0.001, chi-square test, one-tailed). Significant ratios during the delay were observed both in multiunits responding to the two S1s and in multiunits not responding to the two S1s, but more frequently in the former population (28.9% vs. 21.3% in task 1; 38.2% vs. 19.7% in task 6). Similar findings were obtained from the 152 single units isolated from the multiunit recordings, although significant high-/low-load spike-rate ratios during the delay were observed less often (15.1% in task 1; 10.5% in task 6; *Figure 4—figure supplement 1*).

Comparison of the responses in the two tasks revealed that the differences in the spike rate during the delay of high- and low-WM-load trials could not simply be explained by differences in late activity evoked by S1 with frequencies A and B or by differences in preparatory activity for the behavioral responses. Significant high-/low-load spike-rate ratios occurred in only one of the two tasks in the majority of the units and in both tasks in the minority of the units (65% vs. 35% for multiunits and 92% vs. 8% for single units; *Figure 4d* and *Figure 4—figure supplement 1b*). These proportions were significantly different from those expected if the significant spike-rate ratios were only due to differences in late or preparatory activity (i.e., 0% vs. 100%; p<0.001, chi-square test, one-tailed). The occurrence of significant spike-rate ratios in only one of the two tasks was related to the units' best frequencies (for details of the assessment of the best frequency, see Sections Behavioral paradigms and Electrophysiological recordings and data analysis). Significant spike-rate ratios only in task 1 occurred more frequently in units with a best frequency of ~3 kHz (2.67–3.37 kHz) than in other units (26.7% vs. 9.4%; p<0.05, chi-square test, one-tailed). Significant spike-rate ratios only in task 6 occurred more frequently in units with a best frequency of ~1 kHz (0.89–1.12 kHz) than in other units (30.1% vs. 15.4%; p<0.05). Tasks 1 and 6 had reversed WM loads for sequences starting with A and with B. Therefore, the task specificity of the occurrence of significant spike-rate ratios in most units is equivalent to a stimulus specificity, that is, the differences emerged only when either A or B (or the S2-response associations) had to be held in WM. We thus argue that in the majority of units, those displaying the task and stimulus specificity, the differences in the spike rate during the delay of high- and low-WM-load trials were due to differences in WM load. Thus, 30.0% (92/307) of the multiunits and 21.7% (33/152) of the single units in core fields of AC were considered to exhibit WM-related spike activity.

In summary, task-relevant information is stored in core fields of AC by temporary changes of the spike rates of specific neurons. For the majority of multiunits, spike rates increased with increasing WM load (yielding ratios >1) and for the minority spike rates decreased with increasing WM load (yielding ratios <1; *Figure 4c,d*). To scrutinize the responses of the two sub-populations, we averaged the responses from the units with significant ratios >1 or <1 separately for high- and low-WM-load trials. These analyses revealed that in both sub-populations the differences in spike activity between high- and low-WM-load trials persisted throughout the delay (*Figure 4f*).

The following observations also exclude the possibility that the differences in the spike rate during the delay between high- and low-WM-load trials could simply be explained by differences in late activity evoked by S1 with frequencies A or B. In both tasks, significant high-/low-load spike-rate ratios during the delay were observed more often ($p < 0.001$, chi-square test, one-tailed) in multiunit recordings obtained when the monkeys performed the tasks correctly (27.4% in task 1 and 34.5% in task 6) than when they made errors (9.8% and 9.5%) or in the responses to the corresponding sequences recorded when the monkeys did not perform the tasks (5.5% and 7.5%; *Figure 4e*). Significant ratios were also rarely observed in the responses to the corresponding sequences recorded in the untrained naive monkey (0.0% and 3.5%; *Figure 4e*). Furthermore, in additional 15 sessions, monkey L performed tasks 1 and 6 also on sequences where S1 and S2 were separated by the longer delay of 1100 ms. Significant ratios during the final 500 ms of the 1100-ms delay (23.2% and 18.8% of 69 recordings in tasks 1 and 6) were observed slightly, but not significantly more frequently ($p > 0.05$ for tasks 1 and 6, chi-square test, one-tailed) than during the final 500 ms of the 800-ms delay (17.7% and 15.3% of 124 recordings). This result is inconsistent with that expected if significant ratios are due to differences in late activity because such differences would be expected to decay over time.

Two ways of temporarily storing task-relevant information by cortical neurons have been identified: persistent enhancement or suppression of the spike rate during the delay relative to baseline (*Fuster and Jervey, 1982*; *Goldman-Rakic, 1995*; *Wilson et al., 1994*; *Zhou and Fuster, 1996*). We also observed both persistent enhancement and suppression in core fields of AC but enhancement clearly dominated in both tasks, and most prominently in high-WM-load trials. This was demonstrated by computing the ratio between the spike rate during the final 500 ms of the delay and that during the 500-ms period before S1 (*delay/baseline spike-rate ratio*). Delay/baseline spike-rate ratios >1 represent enhancement and ratios <1 suppression. *Figure 4g* shows these ratios, separately for high- and low-WM-load trials, for each of the multiunits with significant high-/low-load spike-rate ratios during the delay (84 in task 1 and 106 in task 6; see above). Delay/baseline spike-rate ratios ranged from 0.5 to 1.6, with the majority of significant ratios occurring in high-WM-load trials (65 of 84 in task 1 and 77 of 106 in task 6) and the minority in low-WM-load trials (21 of 84 in task 1 and 13 of 106 in task 6). In both high- and low-WM-load trials, and in both tasks, the majority of the significant delay/baseline ratios were >1 (*Figure 4g*).

## Origins of working-memory related activity in core fields of auditory cortex

We asked whether WM-related spike activity arose from local processing within core fields of AC or whether other brain structures contributed to it. For this purpose, we analyzed LFPs in core fields of AC, which were simultaneously recorded along with the spike activity in core fields of AC, and spike activity in PFC, which was recorded in separate sessions in monkey C. LFPs in core fields of AC partly reflect synaptic inputs to core fields of AC, and PFC has been hypothesized to exert a top-down modulation of activity in sensory cortex to aid storage of information (*Sreenivasan et al., 2014*).

During the delay, differences in WM load were also reflected in the LFPs recorded in core fields of AC. *Figure 5a and b* show two representative sites where the LFPs during the final 500 ms of the delay of high- and low-WM-load trials were significantly different when the monkeys performed task 1 correctly. These differences emerged shortly after S1 and persisted until the end of the delay. LFP differences were computed by subtracting the amplitude in the low-WM-load trials from that in the high-WM-load trials (*high- – low-load LFP difference*). *Figure 5c* shows the distributions of these differences during the final 500 ms of the 800-ms delay (green dots and traces) and during baseline (black dots and traces) for all 310 sites from both monkeys combined. Similar to the results obtained with spike activity, significant LFP differences occurred more frequently during the delay than during baseline ($p < 0.001$, chi-square test, one-tailed). The frequency of occurrence of significant LFP

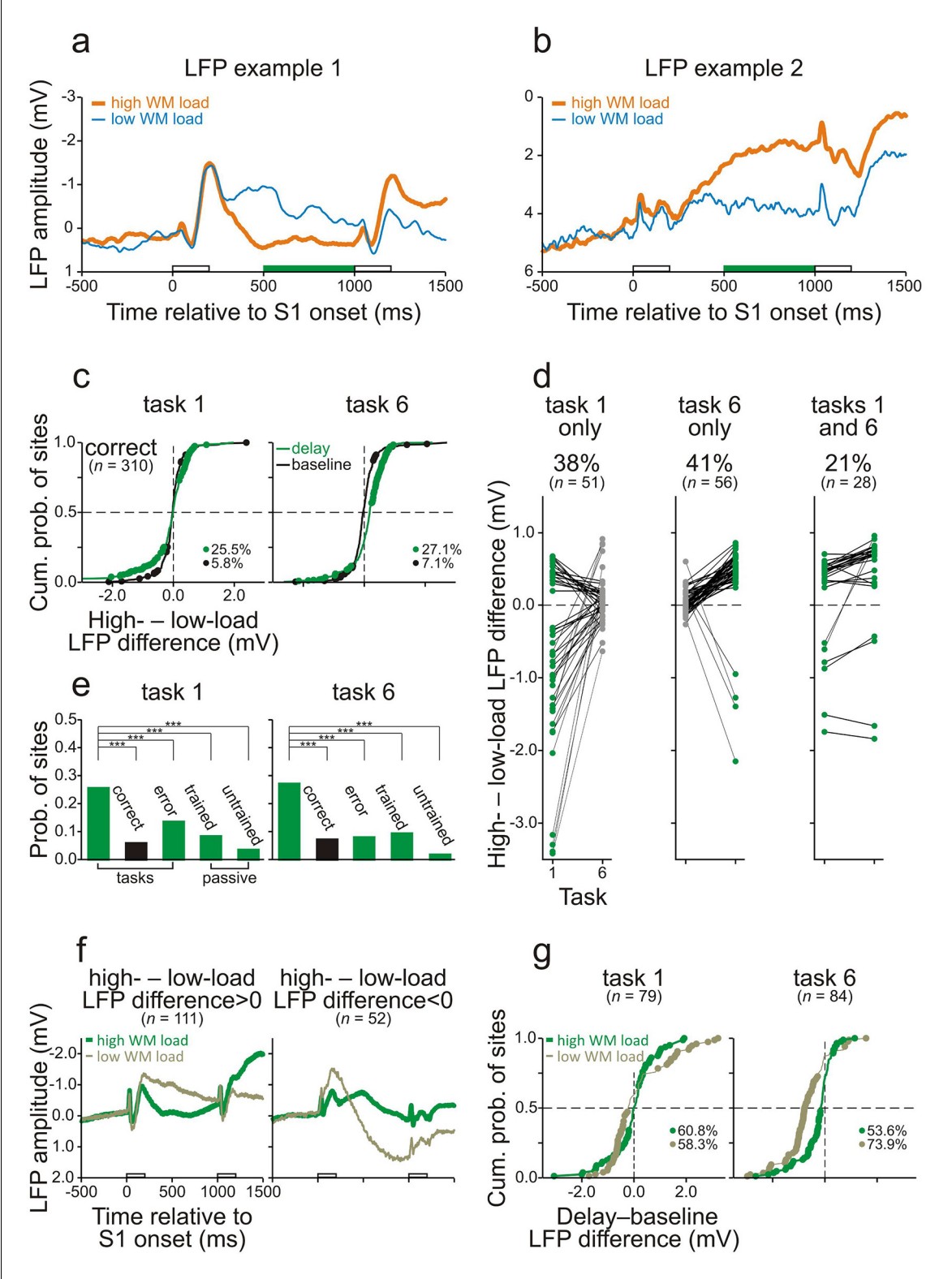

**Figure 5.** Working-memory related potentials in monkey auditory cortex. (**a**, **b**) Representative local field potentials (LFPs) recorded at two different sites in AC of the two monkeys while they performed task 1 correctly. The organization of the figure and other conventions are equivalent to those of *Figure 4*. LFPs were compared by subtracting the LFP amplitude in low-WM-load trials from that in high-WM-load trials (*high- – low-load LFP difference*). In panel g, LFPs were compared by subtracting the mean LFP amplitude during the 500-ms period before S1 (*baseline*) from that during the final 500 ms of the delay (*delay–baseline LFP difference*).

*Figure 5 continued on next page*

*Figure 5 continued*

The following figure supplement is available for figure 5:

**Figure supplement 1.** Scheme of the superposition of the effects of working memory and other mental processes on neural activity.

differences during the delay also depended on the behavioral context, being significantly higher (p<0.001, chi-square test, one-tailed) when the monkeys performed the tasks correctly (25.5% and 27.1% in tasks 1 and 6, respectively) than when they made errors (13.5% and 7.9%) or than in the responses to the corresponding sequences when the monkeys did not perform the tasks in the passive condition (8.3% and 9.3%; *Figure 5e*). Significant LFP differences during the delay were also observed more often when the monkeys performed the tasks correctly than in the responses to the corresponding sequences in the untrained naive monkey (3.5% and 1.7%; *Figure 5e*). Also mirroring the spike activity, significant LFP differences occurred in only one of the two tasks at the majority of sites (107 of the 135 sites with a significant LFP difference in one or both tasks; 79%) and in both tasks at the minority of sites (28 of 135; 21%; *Figure 5d*). Based on these results, we conclude that at 107 of 310 sites (34.5%) the LFP differences between high- and low-WM-load trials were due to differences in WM load. Thus, task-relevant information is also stored in the synaptic potentials in core fields of AC. As with spike activity, we found that LFPs were persistently different during the delay, such that at some sites LFPs were more positive in high- than in low-WM-load trials (left panel in *Figure 5f*), and at other sites more negative (right panel in *Figure 5f*). Both effects were observed at many sites (*Figure 5c,d*), unlike in the case of the spike activity where increases in spike rate with WM load were much more common than decreases (*Figure 4c,d*).

Another difference between spike activity and LFPs was revealed when we examined how LFPs during the delay differed from baseline. *Figure 5g* plots the *delay–baseline LFP differences*, defined as the differences between the mean LFP amplitude during the final 500 ms of the delay and that during the 500-ms period before S1. This measure was calculated for the 79 sites in task 1 and the 84 sites in task 6 with a significant high- – low-load LFP difference. In both tasks, significant delay–baseline LFP differences occurred at many of these sites, both in high- and in low-WM-load trials (60.8% and 58.3%, respectively, in task 1; 53.6% and 73.9% in task 6). *Figure 5g* shows that negative delay–baseline LFP differences prevailed over positive ones in both high- (31.7% vs 29.1% in task 1; 41.7% vs 11.9% in task 6) and low-WM-load trials (38.0% vs 20.3% in task 1; 69.1% vs 4.8% in task 6). At many of the 79 sites in task 1 (41.8%) and the 84 sites in task 6 (67.9%), LFPs during the delay changed more strongly relative to baseline in low- than in high-WM-load trials. It is conceivable that the LFPs at these sites still have the more common relationship to WM load, i.e., change more strongly in high- than in low-WM-load trials. This relationship could be disguised by other mental processes if they had a large effect on LFPs in the direction opposite to that of WM load (*Figure 5— figure supplement 1*). This conjecture also fits the observation that there was a very small number of sites at which the LFPs during the delay changed in opposite directions in high- and low-WM-load trials.

Because differences in WM load were reflected in LFPs, it is possible that synaptic inputs to core fields of AC contribute to the differences in spike activity between high- and low-WM-load trials in core fields of AC. To clarify whether some of these inputs arise from PFC, we also recorded spike activity from the ventrolateral part of PFC (vlPFC) in monkey C (*Figure 1—figure supplement 3*). This part of PFC has been shown to play an essential role in auditory WM tasks (*Hwang and Romanski, 2015*; *Plakke and Romanski, 2014*; *Plakke et al., 2015*). We analyzed 238 multiunits recorded during 70 sessions in which the monkey performed tasks 1 and 6. Like in core fields of AC, neurons in vlPFC could discharge significantly more spikes (p<0.05, permutation test) during the delay of high- than of low-WM-load trials (*Figure 6a*) or vice versa (*Figure 6b*). We observed that, compared to core fields of AC, a similar proportion of multiunits in vlPFC displayed high-/low-load spike-rate ratios during the delay differing significantly from 1. Also, there was a similar dependence of the occurrence of significant high-/low-load spike-rate ratios on the behavioral context (cf. *Figures 6c* and *4c*, *Figures 6e* and *4e*), and a similar task or stimulus specificity (cf. *Figures 6d* and *4d*). These results indicate WM-related activity also in vlPFC. Unlike in core fields of AC, where overall

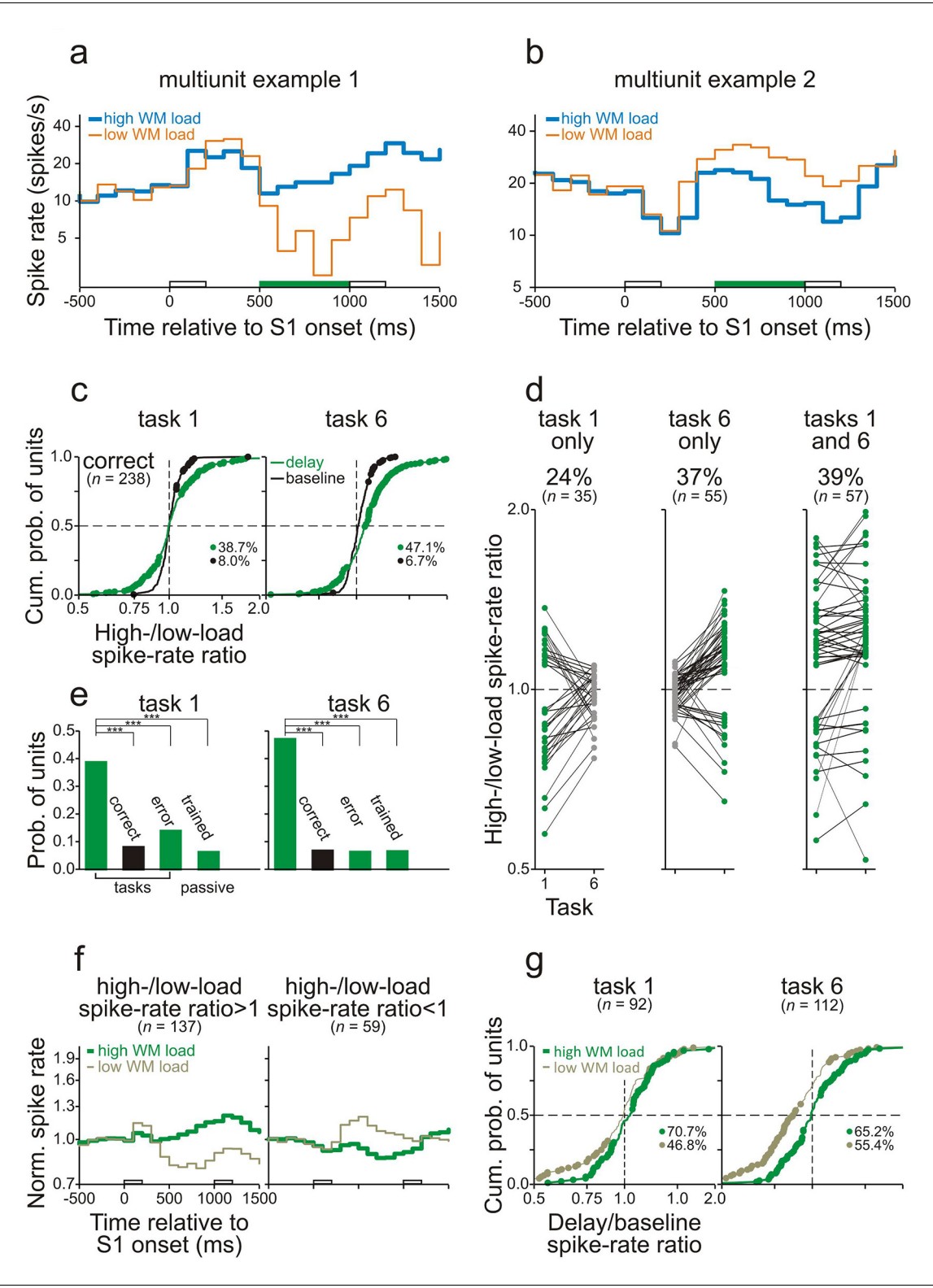

**Figure 6.** Working-memory related spike activity in monkey ventrolateral prefrontal cortex. (a, b) The spike rates of two representative multiunits recorded in vlPFC from monkey C while he performed task 6 correctly. The organization of the figure and all other conventions are identical to those of *Figure 4*.

enhancement during the delay dominated and spike rates predominantly changed in high-WM-load trials (*Figure 4f,g*), in vlPFC suppression was as common as, or even more common than, enhancement during the delay (*Figure 6d,f,g*). Spike rates also frequently changed in low-WM-load trials, and sometimes even more strongly than in high-WM-load trials (*Figure 6f,g*). The conjecture in *Figure 5—figure supplement 1* fits this finding.

Core fields of AC and vlPFC also differed in the time course of WM-related spike activity (cf. *Figure 4f* and *Figure 6f*). This issue was quantified by determining for each multiunit the earliest time, relative to the offset of S1, at which the spike rates during the delay of high- and low-WM-load trials differed significantly (*differential latency*; permutation test, p<0.05, two-tailed). This analysis was conducted in consecutive 10-ms bins and restricted to those multiunits with significant high-/low-load spike-rate ratios during the delay. *Figure 7* shows the distributions of the differential latency of units in core fields of AC (dotted traces) and in vlPFC (solid traces). In both tasks, differential latencies were significantly shorter in core fields of AC than in vlPFC, with a median difference of ~100 ms (p<0.001, permutation test, two-tailed). This finding is difficult to reconcile with the notion that PFC exerts a top-down modulation of activity in sensory cortex (*Sreenivasan et al., 2014*) and thus triggers the storage of information in core fields of AC on a trial-by-trial basis.

## Discussion

The current report provides converging evidence from humans and monkeys that neural activity in AC can reflect WM. This was demonstrated in three independent studies in which subjects performed different tasks on the same sequences of sounds, allowing separation of activity related to WM from activity related to other mental processes. Changes in WM load resulted in persistent changes of spiking activity, LFPs and magnetic fields, suggesting that AC stores information in spikes and synaptic potentials. Our results thus support the idea that early sensory cortex stores information for WM (*Sreenivasan et al., 2014*).

### Approaches to separate working memory from other mental processes

An auditory task has been defined as an operation requiring the production of an intentional connection between sounds, other context-relevant stimuli, and behavioral actions, which are executed because subjects are motivated to do so (*Scheich and Brosch, 2012*). Activity in AC is not only related to processing the sounds and the other context-relevant stimuli but also to mental processes

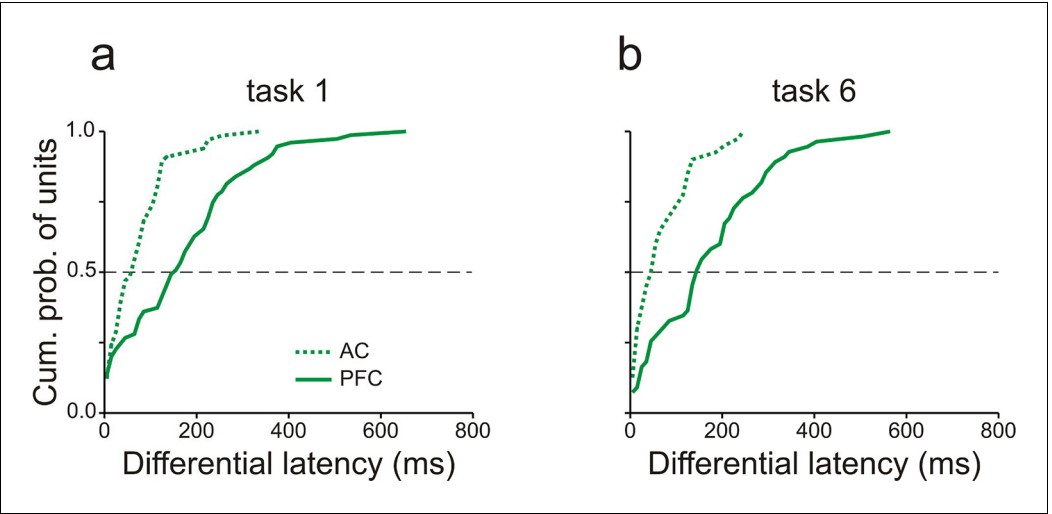

**Figure 7.** Working-memory related spike activity emerges earlier in auditory cortex than in prefrontal cortex. Differential latencies (see Results for definition) of multiunits recorded in AC (dashed traces) and vlPFC (solid traces) in tasks 1 (a) and 6 (b). The dashed horizontal lines mark the median cumulative probability. Data were obtained from monkey C.

associated with performing a task such as selection of and preparation for behavioral actions, and expectation of rewards (*Brosch et al., 2011*). In the current report, we confirmed that activity in AC can persistently change for several seconds relative to baseline in a sound-discrimination task (task 5; *Figure 3*). Thus, persistent changes of neural activity observed during the performance of WM tasks in previous studies are, on their own, not sufficient for demonstrating that AC is involved in WM (*Bigelow et al., 2014*; *Gottlieb et al., 1989*; *Grimault et al., 2014*; *Kumar et al., 2016*; *Lemus et al., 2009*; *Linke and Cusack, 2015*; *Linke et al., 2011*; *Ng et al., 2014*; *Nolden et al., 2013*; *Scott et al., 2014*).

In the current report, we used three approaches to show that at least some portion of persistent changes in activity in AC was related to WM and not to other concurrent mental processes. In the first approach (study 1), we compared differences in activity between trials that differed in WM load in one task with differences in activity between trials that did not differ in WM load in another task. In the second approach (study 3), we compared differences in activity between trials that differed in WM load in one task with differences in activity between trials that also differed in WM load, but in a reversed fashion, in another task. This further enabled the determination of the stimulus-specificity of the effect of WM load. In the third approach (study 2), we directly compared activity in trials which differed only in WM load. Our approaches have the potential to single out other types of WM-related activity including rhythmic activities and spatiotemporal activation patterns and could be used in other sensory modalities. They can also be adapted to tease out activity related to other cognitive and motivational processes. However, note that the second approach requires stimulus-specificity of the neural activity under consideration. It may be difficult to demonstrate such stimulus-specificity in gross measures of the electrical activity of large neural populations, such as EEG and MEG.

## How is information temporarily stored in auditory cortex?

Our converging results from intracranial recordings in monkeys and extracranial recordings in humans demonstrated that information is temporarily stored in AC by persistent changes in spike rates and slow potentials. This was reflected in enhancement or suppression of spike rates, positive or negative changes of LFPs, and increases in the strengths of regional sources during the delay relative to baseline. These ways of storing information in AC are similar to those considered for visual (*Fuster and Jervey, 1982*), somatosensory (*Zhou and Fuster, 1996*), and prefrontal cortex (*Goldman-Rakic, 1995*; *Reinhart et al., 2012*; *Wilson et al., 1994*). Enhancement of activity in AC during the delay may serve to refresh representations of information to keep them active while suppression of activity may serve to protect representations from internal or external interference (*Linke et al., 2011*; *Scott et al., 2014*). The latter has been found to be particularly relevant when recoding and rehearsal strategies were not used by subjects (*Linke et al., 2011*). Information could also be stored in AC in other ways (*Stokes, 2015*), such as in spatiotemporal activation patterns, synaptic plasticity, or rhythmic activities, but these were not analyzed here.

Although our approaches revealed sites where neural activity was related to WM, it is likely that other mental processes contribute to the persistent changes in activity during the delay relative to baseline. In the spike activity, effects of other mental processes seem to be relatively small, because the spike rate barely changed during the delay relative to baseline in low-WM-load trials (*Figure 4f, g*). In the LFPs, effects of other mental processes seem larger, because at many sites LFPs changed during the delay even in low-WM-load trials (*Figure 5g*). This is consistent with the large increase of source strength during the delay relative to baseline in our MEG studies, which was only slightly smaller in low- than in high-WM-load trials (*Figures 2*, *3*). LFPs and magnetic fields have been considered to partially reflect inputs to a region of tissue, in contrast to spike activity, which has been considered to preferentially reflect output to connected areas (*Buzsáki, 2004*; *Buzsáki et al., 2012*; *Hämäläinen et al., 1993*). Our observations that mental processes other than WM were represented differently at the input to and at the output from AC therefore suggest that selective input-output transformations occur in AC such that, at the output (spikes), the representation of WM is favored over that of other mental processes.

## What information is temporarily stored in auditory cortex?

In our experiments, subjects could have used two behavioral strategies to solve the tasks which required WM. They could have stored either the frequency of S1 during the delay or the stimulus-response associations of S2. Which information was stored in an individual trial is difficult to assess and cannot be inferred from the behavioral response in that trial. Therefore, we cannot determine what information is reflected in the neural activity in AC. This question can be addressed in future experiments with tasks that can only be solved with a single strategy. For instance, increasing the number of frequencies for S1 in study 2 would bias the subject to solve the tasks by storing the information of the past S1, as tested in previous studies on animals (e.g., *Gottlieb et al., 1989*).

The WM-related activity in the core fields of AC, as measured in monkeys, was stimulus specific, i. e., it was observed only when a specific frequency of S1 (or specific stimulus-response associations of S2) had to be stored in WM. Since only two tones (1 and 3 kHz) were tested, the stimulus selectivity could not be assessed accurately. Our study confirms the stimulus-specificity of WM-related activity in AC revealed in fMRI studies of humans (*Kumar et al., 2016*; *Linke et al., 2011*) but contrasts with studies of nonhuman primates which reported stimulus-specificity to be missing in AC (*Lemus et al., 2009*; *Scott et al., 2014*).

Our report provides evidence for the sensory-recruitment hypothesis of WM (*Sreenivasan et al., 2014*), which posits that storage of information recruits the same neurons in sensory cortex that represent the information. This hypothesis has mainly been based on multivariate analysis of fMRI signal patterns in early visual cortex, thus, on a signal that is only indirectly related to brain activity (*Logothetis and Wandell, 2004*), but not on direct measurements of electric activity in sensory cortex. Our results are also compatible with the static population coding hypothesis for WM which posits that storage of information recruits neurons without high selectivity (*Sreenivasan et al., 2014*), because we also observed WM-related activity in neurons that did not respond to the two tones (1 and 3 kHz).

## Spatial distribution of working-memory related neural activity

Our results imply that single neurons in AC are capable of temporarily storing information by changing their spike rates. Such neurons were found in the core fields of AC, where they comprised 22% of all recorded neurons. It is conceivable that these neurons are spatially organized in clusters preferentially exhibiting either enhancement or suppression during the delay. This speculation is consistent with our finding that WM-related activity in multiunits was as common as in single units (30% vs. 22%). This would not have been observed if the neurons exhibiting enhancement or suppression were not clustered. Following similar reasoning, spatial clustering may also be present for the slow potentials, otherwise changes of LFPs in positive or negative directions would not have been observed during the delay.

While recordings of spike activity provide clear evidence for WM-related activity in the core fields of AC, we cannot clarify whether other fields of AC are also involved in WM. Because the slow potentials analyzed here can spread over millimeters, it is possible that synaptic potentials in adjacent AC fields contributed to the WM-related LFPs measured in the core fields. An involvement of AC fields outside the core in the performance of WM tasks has been demonstrated for rostral superior temporal cortex (*Scott et al., 2014*) and temporal pole (*Ng et al., 2014*). Our MEG studies clarified that activity in both the left and right AC was related to WM. Involvement of both ACs has been seen in previous studies for simple WM tasks with other stimuli (*Grimault et al., 2014*; *Kumar et al., 2016*; *Nolden et al., 2013*), for complex WM tasks that, e.g., required manipulations of WM content (*Brechmann et al., 2007*), or for WM tasks in which subjects utilized other strategies, such as articulatory rehearsal (*Linke et al., 2011*).

Our report also revealed WM-related activity in vlPFC. Such activity is consistent with previous studies showing that vlPFC is involved in non-spatial auditory WM tasks (*Hwang and Romanski, 2015*; *Plakke and Romanski, 2014*; *Plakke et al., 2015*). Although based on one monkey only, our observation of persistent enhancement and suppression of spike rates during the delay suggests that vlPFC stores information in the same ways as AC. Our observation that WM-related activity started ~100 ms later in vlPFC than in AC suggests that vlPFC does not trigger the storage of information in AC on a trial-by-trial basis. Although PFC has been traditionally hypothesized to exert a top-down modulation to aid storage of information in sensory cortex (*Curtis and D'Esposito, 2003*;

*D'Esposito and Postle, 2015*; *Fuster, 2001*; *Miller and Cohen, 2001*; *Miller et al., 1996*; *Sreenivasan et al., 2014*), our finding challenges vlPFC as the source of top-down signals to auditory cortex.

### Role of working-memory related activity in auditory cortex for hearing

The present study reported a sizable number of neurons in AC with activity related to auditory WM. The storage in AC of information about sounds over several seconds may be an essential component of the analysis of temporal patterns of sounds including speech and communication sounds (*May et al., 2015*; *Petkov and Jarvis, 2012*). The WM-related persistent activity changes described here reflect active processes recruited when subjects are engaged in WM tasks. Because the change of the neural activity persisted throughout most of the period during which information needed to be stored, it is conceivable that the memory trace carried by this activity remains relatively unaltered throughout the memory period. Whether persistent activity changes in AC also support the storage of information for even longer periods and other WM processes, including spatial aspects and processes requiring modifications of stored information, is not clear and is under debate (*Fritz et al., 2005*; *Scott and Mishkin, 2016*). Because very similar persistent WM-related activity was observed in our studies in the AC of humans and monkeys, the cellular bases of some auditory WM processes in humans can well be studied in monkeys.

## Materials and methods

### Human studies

#### Behavioral paradigms

Fifteen subjects (7 males, 8 females, mean age = 26 years) participated in study 1 and twenty-six (14 males, 12 females, mean age = 27 years) in study 2. All subjects gave their written informed consent to participate in the studies which were approved by the ethics committee of the Otto-von-Guericke University, Magdeburg. Data from three subjects in study 1 and from 11 subjects in study 2 were excluded prior to analysis because of strong artifacts or technical problems during measurements. The studies were designed and conducted following the guidelines for basic MEG research in the field of neuroscience (*Gross et al., 2013*).

For study 1, subjects were instructed to perform tasks 1, 2 and 3 on an identical set of four two-tone sequences (AA, AB, BA and BB) within the same experimental session. Each trial started with a soft noise burst (a 25-ms burst of white noise with a sound pressure level of 16 dB), accompanied by the appearance of a fixation cross on a screen ~1 m in front of the subject's head. One second later, the first tone (S1) was presented followed by the second tone (S2) two seconds later. Both tones had durations of 100 ms. The delay between S1 and S2, defined as the time from the offset of S1 to the onset of S2, was therefore 1.9 s. The frequencies of the two tones were either A (1.5 kHz) or B (1.6 kHz) and their sensation levels were adjusted to 80 dB SL. Stimuli were conducted to the subject's ear via a plastic tube and ear mold which fitted comfortably into the external ear canal. When a go response was required, as explained below, subjects had to make it within $2 \pm 0.25$ s after the onset of S2 when the next trial started (see *Zacharias et al., 2011* for details of the experimental setup).

For tasks 1 and 2 (*Figure 1a*), subjects were instructed to make a go response (i.e., press a response button) when both S1 and S2 were A and a nogo response (no button press) for the other three sequences (AB, BA and BB). Sequences starting with A were high-WM-load trials and those starting with B were low-WM-load trials. Tasks 1 and 2 differed only with respect to the ear of stimulation and the hand used to make the response. For task 1, stimuli were conducted to the subject's left ear and the go response was made with the right index finger, contralateral to the stimulated ear. For task 2, stimuli were conducted to the right ear and the go response was made with the left index finger, also contralateral to the stimulated ear. For task 3 (*Figure 1c*), subjects were instructed to make a go response when S1 was A and a nogo response when S1 was B, irrespective of the frequency of S2. As for task 1, the stimuli were conducted to the left ear and the right index finger was used for the go response. Each subject performed the three tasks in three separate blocks. The order of the tasks was randomized across subjects.

For study 2, subjects were instructed to perform tasks 4 and 5 on two different sets of two-stimulus sequences which differed in S2 but not in S1. All stimuli had durations of 100 ms and were presented at a sound pressure level of 80 dB. Task 4 (*Figure 1d*) was a delayed-match-to-sample task, for which the same three tones were used as S1 and S2 (A, B and C with frequencies of 1.5, 0.789 and 2.85 kHz, respectively). The frequencies of B and C were selected such that the ratios A/B and C/A were equal to 1.9. Subjects had to make a go response when S1 and S2 were identical (AA, BB, CC) and a nogo response when S1 and S2 were different (AB, AC, BA, BC, CA, CB). For task 5 (*Figure 1e*), the same three tones were used as S1, whereas S2 was different. S2 was either a uniform-white-noise burst (D), a tone modulated in frequency from 0.789 to 2.850 kHz (E), or a train of 25 clicks of 1-ms duration each and presented at a rate of 250 clicks per second (F). Regardless of S1, subjects had to make a go response when S2 was the noise burst (AD, BD, CD), and a nogo response when S2 was the frequency modulated tone or the click train (AE, BE, CE, AF, BF, CF). All sequences in task 4 were high-WM-load trials and those in task 5 were low-WM-load trials.

For both tasks, the delay between S1 and S2 was 1.9 s. Subjects had to make their go responses within 1.5 s after the onset of S2. Each trial started with a visual cue (a fixation cross). A visual feedback was provided immediately after the subject's response (a smiling face for a correct response and a frowning face for an incorrect response). The inter-trial interval was 2 s. Stimuli were conducted to the subject's left ear and go responses were made with the right index finger, contralateral to the stimulated ear. Each subject performed the two tasks in two separate blocks within one experimental session. The order of the tasks was randomized across subjects.

## MEG recordings, data preprocessing and analysis

MEG measurements were performed in a magnetically shielded chamber by means of a 248-channel whole-head magnetometer system (4-D Neuroimaging, San Diego, USA). The magnetic field was acquired continuously at a sampling rate of 678.17 Hz in study 1 and of 1017.25 Hz in study 2. Signals were filtered online between DC and 100 Hz in study 1 and between DC and 200 Hz in study 2. Note that high-pass filtering signals at 4 Hz revealed no differences between high- and low-WM-load trials. In addition, vertical and horizontal electrooculograms were recorded to track eye movements and blinks.

For further processing, the raw data were exported into the BESA software package (BESA Software GmbH, Gräfelfing, Germany). We first performed a heartbeat artifact correction (*Ille et al., 2002*). The corrected signal was then segmented into 3-s epochs, ranging from 500 ms before the onset of S1 to 500 ms after the onset of S2. Epochs contaminated with other artifacts were discarded (for details, see *Zacharias et al., 2011*). For each subject, task and sensor, the remaining artifact-free epochs were band-pass filtered (high-pass DC, low-pass 30 Hz, zero phase, 24 dB/octave). Average waveforms were obtained by arithmetically averaging the signals from epochs of trials with correct responses, separately for each subject, task and sensor. The use of the arithmetic average for this purpose is justified (*Zacharias et al., 2011*; *König et al., 2015*). The average waveforms were then baseline-offset corrected. The value for the baseline offset correction was determined from the mean signal amplitudes over the 300-ms interval directly before S1 and was subtracted from the waveform.

The baseline-corrected average waveforms from all sensors formed the input to the source analysis for which each dataset was co-registered with the T1-weighted anatomical MR image (3T, 192 slices) of the subject's brain using Brainvoyager QX (Brain Innovation B.V., Maastricht, The Netherlands). Separately for each subject and task, we modeled the neuronal sources underlying the magnetic field distribution around the peak of the M100 component evoked by S1 by two regional sources, one in the left and one in the right AC. These regional sources were seeded at the border of Heschl's gyrus and planum temporale (*Näätänen and Picton, 1987*). In addition, we seeded three pairs of regional sources roughly symmetrically in left and right central, frontal and occipital cortical areas (*Figure 1—figure supplement 1*) to model background activity, that is, to exclude potential confounds from other brain areas. These three pairs of sources were located close to the cortical surface such that their distances to the source in the AC of the same hemisphere were relatively large. We did so to minimize the mutual influence between the signals of these sources and of the sources in AC because the mutual influence decreases with increasing distance between the sources. In contrast to the positions of the sources, their orientations and amplitudes were fitted to best account

for the magnetic field distribution of the average waveforms at all sensors using the spherical volume conductor model as head model (*Scherg, 1990*). Results of this source analysis are source waveforms, one for each subject and condition. These waveforms were then exported to MATLAB (Math-Works, Natick, MA) for further analysis.

The comparison of grand mean waveforms across tasks requires homogeneous variance and a normal distribution of the underlying MEG data. We found that this could be achieved by a log-transformation of individual source waveforms, consistent with earlier studies (*König et al., 2015*; *Matysiak et al., 2013*; *Zacharias et al., 2011*). To obtain the grand mean waveform, the log-transformed source waveforms were arithmetically averaged across subjects and this mean waveform was then back-transformed (by exponentiation) and plotted along a logarithmic axis.

Statistical analyses were performed by permutation tests. The significance level was set to 0.05.

## Monkey study

### Behavioral paradigms

Three macaque monkeys (Macaca fascicularis) participated in this study. Two of them (monkeys C and L) were trained to perform tasks 1 and 6, on an identical set of four two-tone sequences (AA, AB, BA and BB) within the same experimental sessions. Task 1 was equivalent to task 1 performed by human subjects. Each trial started with the illumination of a LED, which signaled that the monkeys had to grasp a bar within 4 s. Shortly after grasping the bar (1–2s), two tones S1 and S2 were presented. Both tones had durations of 200 ms. The stimulus-onset interval between S1 and S2 was 1000 ms in most sessions, resulting in an 800-ms delay, defined as the interval from the offset of S1 to the onset of S2. In other sessions, the stimulus-onset interval between S1 and S2 was 1300 ms, resulting in an 1100-ms delay. The frequencies of tones A and B were 3 kHz and 1 kHz, respectively, and their sound pressure level was 67 dB SPL. Each tone was presented through two loudspeakers (Karat 720), located on the left and right side of the monkey. When a go response was required, monkey L had to release the bar within 240–1360 ms after the onset of S2, and monkey C within 240–1960 ms. When a nogo response was required, the monkeys had to continue holding the bar beyond the end of these time windows. Upon correct responses, the monkeys were rewarded with a drop of water (0.2–0.3 ml). The monkeys were given at least 5 s for drinking water before the start of the next trial. The monkeys performed the tasks using their left hand.

For task 1 (*Figure 1a*), the monkeys had to make a go response when both S1 and S2 were A and a nogo response for the other three sequences (AB, BA and BB). For task 6 (*Figure 1b*), the monkeys had to make a go response when both S1 and S2 were B and a nogo response for the other three sequences (BA, AB and AA). In each experimental session, the monkeys performed the two tasks in separate, alternating blocks. In each block, trials requiring a go or nogo response were presented in random order, with go responses in ~60% of the trials and nogo responses in ~40% of the trials. In task 1, the LED was green and located on the right side. In task 6, the LED was red and located on the left side.

The same set of stimuli was also presented in blocks to the two trained monkeys and to a third untrained naïve monkey (monkey M) when they did not perform the tasks in the way that the LEDs were not illuminated and the touch bar was not functional. There were blocks in which the AA sequence occurred in ~60% of the trials and the other three sequences ~40% of the trials, as in task 1. In other blocks, the BB sequence occurred in ~60% of the trials and the other three sequences ~40% of the trials, as in task 6. For the two trained monkeys, these passive-condition measurements were made after they had performed the behavioral tasks, or on days in which they did not perform the behavioral tasks at all.

In an additional passive block, pure tones with 40 different frequencies were presented at ~60 dB SPL to the monkeys to assess each unit's best frequency. Frequencies were equidistantly spaced on a logarithmic scale over a range of 8 octaves (0.0625–16 kHz). Tone duration was 100 ms. Each tone was repeated 10 times. All tones were presented in a pseudorandom order with a stimulus-onset interval of 500 ms. These tones were presented before the monkeys performed the behavioral tasks, or before the passive-condition measurements on days in which the monkeys did not perform the tasks.

The experiments were approved by the authority for animal care and ethics of the federal state of Saxony-Anhalt (No. 28-42502-2-1129IfN), and conformed to the rules for animal experimentation of the European Communities Council Directive (86/609/EEC).

## Electrophysiological recordings and data analysis

For simultaneous recordings of action potentials and local field potentials (0.1–140 Hz) from right AC core area (mainly from A1) and left ventrolateral prefrontal cortex (vlPFC; *Figure 1—figure supplement 3*), we used a seven-channel microelectrode manipulator (2–2.5 MΩ; for details, see *Brosch et al., 2005*). By means of the built-in spike detection tools of the data acquisition systems (threshold crossings and spike duration), we discriminated the action potentials of a few neurons (multiunit) from each electrode and stored the time stamp and the waveform of each action potential. Single unit activity was extracted off-line from the multiunit recordings by means of principal component analysis. The position of A1 was estimated using the spatial distribution of the units' best frequencies and the penetration traces of the electrodes, i.e., electrodes passed through parietal cortex into the auditory cortex (*Brosch et al., 2005*; *Kaas and Hackett, 2000*). The vlPFC was identified by its anatomical location (below the principal sulcus and anterior to the arcuate sulcus; *Romanski and Goldman-Rakic, 2002*) and by the latency and selectivity of neuronal responses for complex auditory stimuli (animal and human vocalizations, natural sounds, man-made sounds, and white noise). This follows the observation of *Romanski and Goldman-Rakic (2002)* that auditory neurons were tightly clustered in a small region of vlPFC outside of which no auditory neurons were found. We also found that neuronal activity in the vlPFC could be synchronized with that in auditory cortex, following the observation of *Romanski et al., (1999)* that the vlPFC reciprocally connected with auditory cortex.

Data analysis was conducted off-line using MATLAB (MathWorks, Natick, MA). For each unit or at each site, statistical analyses were performed by permutation tests of the neuronal activity during the final 500 ms of the delay. High-pass filtering the local field potentials at 4 Hz revealed no differences between high- and low-WM-load trials. For each unit, the best frequency was assessed by analyzing the responses to the 40 pure tones as described by *Brosch et al., (1999)*.

## Acknowledgements

We thank Norman Zacharias, Nadia Abu Eid, and Aida Hajizadeh for assistance in MEG data collection and analysis, Cornelia Bucks for technical assistance of monkey experiments, and Drs. André Brechmann, Patrick May, Dexter Irvine, Erich Schröger and Brian Scott for their valuable comments on the manuscript. We also acknowledge the support by an Alexander von Humboldt Polish Honorary Research Scholarship by the Foundation for Polish Science. This research was supported by a LIN Special Project to PH, RK, and MB, and by the Deutsche Forschungsgemeinschaft (He 1721/10-1, He 1721/10-2, and SFB TR31, A4).

## Additional information

### Funding

| Funder | Grant reference number | Author |
| --- | --- | --- |
| Deutsche Forschungsgemeinschaft | He 1721/10-1 | Peter Heil<br>Reinhard König<br>Michael Brosch |
| Deutsche Forschungsgemeinschaft | He 1721/10-2 | Peter Heil<br>Reinhard König<br>Michael Brosch |
| Deutsche Forschungsgemeinschaft | SFB TR31 | Peter Heil<br>Michael Brosch |
| LIN special project | | Peter Heil<br>Reinhard König<br>Michael Brosch |

The funders had no role in study design, data collection and interpretation, or the decision to submit the work for publication.

### Author contributions

YH, AM, Conception and design, Acquisition of data, Analysis and interpretation of data, Drafting or revising the article, Contributed unpublished essential data or reagents; PH, RK, MB, Conception and design, Analysis and interpretation of data, Drafting or revising the article

### Author ORCIDs

Ying Huang, ⬤ http://orcid.org/0000-0002-6471-8009

### Ethics

Human subjects: All subjects gave their written informed consent to participate in the studies which were approved by the ethics committee of the Otto-von-Guericke University, Magdeburg.
Animal experimentation: The experiments were approved by the authority for animal care and ethics of the federal state of Saxony-Anhalt (No. 28-42502-2-1129IfN), and conformed to the rules for animal experimentation of the European Communities Council Directive (86/609/EEC).

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
