## [Decision Letter]

Thank you for submitting your article "Persistent neural activity in auditory cortex is related to auditory working memory in humans and nonhuman primates" for consideration by *eLife*. Your article has been favorably evaluated by David Van Essen (Senior editor) and three reviewers, one of whom is a member of our Board of Reviewing Editors. One of the reviewers involved in the assessment of your article has agreed to reveal his identity: Mark D'Esposito (Reviewer #3).

The reviewers have discussed the reviews with one another and the Reviewing Editor has drafted this decision to help you prepare a revised submission.

Summary:

The aim of this study was to investigate whether working memory is encoded in auditory cortex by using MEG in humans and extracellular recordings of spiking activity and local field potentials in monkeys. The authors go to great lengths to rule out a contribution of other factors, specifically persistent sound-driven activity and activity preparatory to making a motor response. Furthermore, the combination of recordings from human and non-human primates is a strength of the study. This manuscript therefore provides evidence of working memory encoding within sensory cortex, corroborating other recent findings and indicating that the temporary storage of information likely recruits the same brain areas that process this information.

Essential revisions:

Although the reviewers saw considerable merit in this study and commended the ambitious nature of the experiments, a number of concerns were raised that need to be considered to bolster the conclusions being drawn.

1) Given that there have been numerous studies that have studied auditory working memory (many of which the authors cite), the authors should be more clear that *none* of these studies have demonstrated persistent neural activity in primary auditory cortex during working memory that was not confounded by other potential processes. The authors state that this is the case, but they should back this up with a brief description of a few representative studies they have cited and why they were not conclusive. This would support their notion that the data presented in this paper represent a definitive advance rather than a replication of previous findings.

2) The number of control task conditions designed to rule out "non-memory" contributions to delay period activity is impressive. However, it is the opinion of the reviewers that this endeavor has only been partially successful, and that the potential contribution of other factors, as set out in the following, needs more consideration in the manuscript.

First, the use of only two stimuli in several of the conditions, generally requiring one of two responses, may activate circuits more related to discrimination learning than to stimulus-specific encoding. In other words, the two sound pairs, AA and AB, could be learned signals. Please address this and whether your human MEG studies can measure activity in the auditory thalamus, which, in addition to the auditory cortex, is typically involved in discrimination learning. Although all of the conditions used the same sounds, it might be that those that required one of two responses depending on the second stimulus might be more heavily processed by particular brain regions.

Working memory load does not necessarily have to be higher during the delay (“the WM load during the delay was higher in trials when S1 was A than when it was B”) if a learned rule is applied after hearing stimulus A. Thus, if stimulus A means do X if A is heard next or do Y if B is heard next, then subjects only need to remember to apply this rule, not necessarily to hold stimulus A in short-term memory. If additional stimuli are used with more pairings as in some of the other controls and the rule of same or different needs to be applied between S1 and S2, then working memory is required. However, this is not the case in the human conditions using two stimuli only or in the non-human primate studies.

Second, the authors need to address the concern that the neuronal activity in the 500 ms immediately preceding the S2 presentation could be a build-up of attention or anticipation/timing of S2 based on the S1 cue. This does not require that the activity be related to working memory. When longer delays were used in the Bigelow et al., 2014, paper with much longer delays, the majority of delay active units showed decreased (some increased) activity in the bins (up to 1000 ms before S2 and strongly in the last 500 ms before S2 presentation), with only a few units showing full delay effects. A ramp-up effect occurred right before S2, similar to panels A, B, C, D, etc. in Figure 2.

Attention, as well as working memory may also vary between conditions 1 and 6. Various predictable and unpredictable ISIs would help to control for that. Similarly, attention to S2 could be very different in task 4 than in task 5 (section 2.2.2).

The finding that the M100 amplitude was the same following S1 in tasks 4 and 5 (section 2.2.2) is not surprising as you would expect attention to be the same regardless of whether high or low working memory load is going to be needed on that trial and this does not rule out that the later activity is attention related.

Section 2.3.2, fourth paragraph: Were these significant ratios increases or decreases? Also, here, in the trained monkey L who performed tasks 1 and 6 on sequences where S1 and S2 were separated by longer delays of 1100 ms, the final 500 ms showed even more units that were significant. This could mean that there was more buildup of anticipation for the upcoming S2 as the animal (neurons) had to wait longer for the S2 presentation.

Section 2.3.1, third paragraph: Please provide the data.

To help answer the question of whether anticipation/timing and attention contribute to the sustained delay activity or whether there is a ramping (up or down) of activity in anticipation of a meaningful S2, the reviewers recommend that you compare in the monkey studies the sessions which used ISIs of 800 ms versus those of 1100 ms.

3) Section 2.1.2, the sentence starting "However, the difference in the right AC…". If this difference was found beyond the final 500 ms of the delay, it may be influenced by the response or reward of being told one is correct. The conclusion that this reflects differences in working memory load does not seem well justified.

4) It is unclear why a given neuron (or set of neurons) should exhibit working-memory related activity only in task 1 or in task 6, unless that is related to the frequency preferences of the neurons. Unfortunately, no information is provided about that. The frequencies of the two tones used in the monkey experiments were 1 and 3 kHz. It is stated in the Discussion (end of section 3.3) that not all neurons responded to these frequencies, but it is essential to provide the best or characteristic frequencies of the neurons in the Results section. Without this, the suggestion in the last paragraph of section 3.3 that the same neurons represent and store information becomes much weaker.

5) No statistics appear to have been applied to the MEG data to test differences between high and low working memory loads, or any task-related signal vs. baseline. In the Results section, it is stated that "sources in the left and right AC were significantly stronger in high than in low WM load trials", but no statistics are presented to back this up.

6) It should be made clear what the authors mean by "AC" or auditory cortex. Are they referring to primary auditory cortex? If so, they should not use this term for the MEG data since one cannot unequivocally localize MEG signals to a definitive cortical location. A similar issue is related to definitely stating that the source closest to the motor cortex is actually motor cortex. The most likely regions that would exhibit preparatory activity are premotor areas. Why didn't the authors attempt to perform a better source location for these regions based on anatomical images in a manner that was done for auditory cortex? Even in the monkey recordings, some justification for the claim that the recordings were localized to "core area (mainly from A1)" is required.

7) The MEG data from task 5 is very perplexing given that persistent activity above baseline is found during a delay period when no working memory is required. Can the authors provide any insight into this finding?

8) In several human fMRI studies (e.g. Curtis, J. Neuro, 2004), delay period activity has been linked to behavior. Have the authors made any attempt to relate activity to behavior in either of their human or monkey data?

9) It is suggested in the first paragraph of section 3.4 that because working memory related activity was observed in a similar proportion of multiunits and single units, such neurons are likely to be organized in clusters. This is an extremely weak argument. Conclusions about local clustering can only be made if neighboring recordings sites show similar effects, whereas those located further apart do not.

10) The comparison between auditory cortex and vlPFC is potentially interesting, particularly given the surprising implication that the effects observed may not originate in PFC. However, this is based on one monkey only and so these findings are arguably premature. Since simultaneous recordings were made from these regions, was there any evidence for functional connectivity between them? If not, how can the authors be sure that they were recording in appropriate regions for PFC to precede auditory cortical activity? At very least, a recording placement figure for vlPFC and AC would add value to the manuscript for comparison to data from other labs. Is the analysis of differential latencies in vlPFC and AC based only the 500 msec preceding S2, or was the entire delay including the S1 offset analyzed? No statistics are provided in the fourth paragraph of section 2.3.3. Even if this is based on permutation tests, as mentioned in the methods, the details should be provided here.

---

## [Author Response]

*Essential revisions:*

*Although the reviewers saw considerable merit in this study and commended the ambitious nature of the experiments, a number of concerns were raised that need to be considered to bolster the conclusions being drawn.*

1) Given that there have been numerous studies that have studied auditory working memory (many of which the authors cite), the authors should be more clear that none of these studies have demonstrated persistent neural activity in primary auditory cortex during working memory that was not confounded by other potential processes. The authors state that this is the case, but they should back this up with a brief description of a few representative studies they have cited and why they were not conclusive. This would support their notion that the data presented in this paper represent a definitive advance rather than a replication of previous findings.

We are convinced that the data presented in our manuscript represent a definitive advance because none of the previous studies has unequivocally demonstrated that persistent neural activity in auditory cortex reflects WM. To make this clearer, we added the following: ‘For example, in some of the imaging (Kumar et al., 2016; Linke and Cusack, 2015) and recording studies (Bigelow et al., 2014; Gottlieb et al., 1989; Scott et al., 2014), persistent changes in activity were revealed by comparing activity during the WM period with that during a baseline period. […] Therefore, differences in the persistent activity revealed in these studies could reflect differences in activity evoked by different stimuli rather than differences in WM load.’

*2) The number of control task conditions designed to rule out "non-memory" contributions to delay period activity is impressive. However, it is the opinion of the reviewers that this endeavor has only been partially successful, and that the potential contribution of other factors, as set out in the following, needs more consideration in the manuscript.*

*First, the use of only two stimuli in several of the conditions, generally requiring one of two responses, may activate circuits more related to discrimination learning than to stimulus-specific encoding. In other words, the two sound pairs, AA and AB, could be learned signals. Please address this and whether your human MEG studies can measure activity in the auditory thalamus, which, in addition to the auditory cortex, is typically involved in discrimination learning. Although all of the conditions used the same sounds, it might be that those that required one of two responses depending on the second stimulus might be more heavily processed by particular brain regions.*

Working memory load does not necessarily have to be higher during the delay (“the WM load during the delay was higher in trials when S1 was A than when it was B”) if a learned rule is applied after hearing stimulus A. Thus, if stimulus A means do X if A is heard next or do Y if B is heard next, then subjects only need to remember to apply this rule, not necessarily to hold stimulus A in short-term memory. If additional stimuli are used with more pairings as in some of the other controls and the rule of same or different needs to be applied between S1 and S2, then working memory is required. However, this is not the case in the human conditions using two stimuli only or in the non-human primate studies.

We agree with the reviewers that in tasks 1, 2, and 6, in which only two tones (A and B) were used as S1, subjects could have used a strategy which the reviewers refer to as ‘discrimination learning’, or a strategy which requires holding relevant features of S1 in working memory. Indeed, we referred to the same two strategies in our original manuscript but used different terminologies, namely, the strategy of storing stimulus-response associations of S2 and the strategy of storing the frequency of S1. We would like to emphasize again that also with the ‘discrimination learning’ strategy, subjects needed to utilize working memory after S1. This is so because they would have had to remember and update rules on a trial-by-trial basis or, using the reviewers’ terms, ‘remember to apply’ specific rules after specific S1s. For example, in task 1, when S1 was A, the rule to be remembered was to make a go response if S2 was A and a nogo response if S2 was B, which comprised two stimulus-response associations. When S1 was B, the rule to be remembered was to make a nogo response, a single stimulus-response association independent of the identity of S2. Therefore, trials where S1 was A and trials where S1 was B differed in working memory load during the delay between S1 and S2. We thus respectfully disagree with the reviewers' concern that 'working memory load does not necessarily have to be higher during the delay if a learned rule is applied'. To make these concepts clearer to readers, we made the following changes in the manuscript.

In the Introduction, we added: ‘The two types of information have been termed the retrospective and prospective codes, respectively, and both can be stored in WM to bridge sensory events or their contingent behavioral actions (Curtis et al., 2004; D’Esposito, 2007; Postle, 2006; Sreenivasan et al., 2014).’

In Section 2.1.1, we added: ‘Irrespective of the strategies subjects utilized, WM was involved during the delay between S1 and S2 in each trial. The random presentation of the four sequences, AA, AB, BA, and BB, prevented the predictability of the sequence in the upcoming trial and, consequently, the information required for making a correct response needed to be stored and updated on a trial-by-trial basis, that is, in WM on a scale of seconds (Goldman-Rakic, 1995).’

In Section 2.1.1, we changed ‘Therefore, the WM load during the delay was higher in trials when S1 was A than when it was B’ to: ‘Because more information needed to be stored during the delay in trials when S1 was A than when it was B, the WM load during delay was higher in the former than in the latter trials.’

In Section 2.3.1, we added: ‘(for detailed reasoning, see the second paragraph in Section 2.1.1)’.

Our experiments do not allow measuring activity in auditory thalamus using MEG. The neuronal currents in auditory thalamus form roughly symmetric patterns (closed fields). The associated magnetic fields decay rapidly as a function of distance and, thus, are very small at the location of the MEG sensors. To detect these tiny signals in the presence of unavoidable background signals, a large number of trials are required, along with special filter settings (bandpass filters between ~150 and 1000 Hz) (for details, see: Attal et al., Rev. Neurosci. 23: 85-95, 2012; Parkkonen et al., Hum. Brain Mapp. 30: 1772-1782, 2009). For example, even after averaging up to 16,000 trials in the study by Parkkonen et al. (2009), responses from auditory thalamus were not apparent. Relating this large number of trials to one of our paradigms means that each subject would have to be measured for at least ~70 hours (~5s per trial × 16,000 trials × 3 tasks), which is not practical.

*Second, the authors need to address the concern that the neuronal activity in the 500 ms immediately preceding the S2 presentation could be a build-up of attention or anticipation/timing of S2 based on the S1 cue. This does not require that the activity be related to working memory. When longer delays were used in the Bigelow et al., 2014, paper with much longer delays, the majority of delay active units showed decreased (some increased) activity in the bins (up to 1000 ms before S2 and strongly in the last 500 ms before S2 presentation), with only a few units showing full delay effects. A ramp-up effect occurred right before S2, similar to panels A, B, C, D, etc. in Figure 2.*

*Attention, as well as working memory may also vary between conditions 1 and 6. Various predictable and unpredictable ISIs would help to control for that. Similarly, attention to S2 could be very different in task 4 than in task 5 (section 2.2.2).*

The finding that the M100 amplitude was the same following S1 in tasks 4 and 5 (section 2.2.2) is not surprising as you would expect attention to be the same regardless of whether high or low working memory load is going to be needed on that trial and this does not rule out that the later activity is attention related.

Working memory and attention have traditionally been viewed as distinct cognitive concepts. However, a growing number of psychological and neuroscientific studies have revealed extensive overlap between these two concepts, inspiring numerous reviews and discussions about the relationship between working memory and attention as well as theories of working memory. One prevailing view is that working memory and attention share the same mechanism of prioritization of processing. Prioritization of processing is traditionally considered a hallmark of working memory when it is directed toward internal representations, and of selective attention when directed toward external stimuli (e.g., Kiyonaga and Egner, 2013; Gazzaley and Nobre, 2012). Recent influential theories of working memory directly consider working memory as internal attention sustained over time and assert that selective attention is the mechanism by which information is stored in working memory (e.g., Awh and Jonides, 2001; Chun, 2011; Postle, 2006). Although these theories have been based mainly on visual studies, evidence has also emerged from auditory studies (for example, see: Backer and Alain, J. Exp. Psychol. Hum. Percept. Perform. 38: 1554-1566, 2012; Backer et al., J. Neurosci. 35: 1307-1318, 2015; Green and McKeown, Percept. Psychophys. 69: 942-951, 2007). A few studies have shown an interplay between attention and auditory memory in the way that attention can be directed to memory traces and that existing memory can guide attention (for an example review, see Zimmermann et al., 2016).

According to these views and theories of working memory, the differences in neural activity during the delay between high- and low-WM-load trials in tasks 1, 2, and 6, and also between trials in tasks 4 and 5, are related to attention because subjects need to allocate more attention in high- than in low-WM-load trials to achieve the storage of information in working memory. Whether WM-related neural activity is ramping up or ramping down or time-invariant is still not clear in auditory cortex because none of the previous studies, including the study by Bigelow et al. (2014), has unequivocally demonstrated persistent activity related to WM. To emphasize the view that attention is a key component of working memory rather than a separate confounding process that can be disentangled from working memory, we made the following changes in our manuscript.

In the Introduction, we added: ‘Research on WM has promoted the view that the storage is accomplished by sustained attention to internal representations of information (Awh and Jonides, 2001; Chun, 2011; Gazzaley and Nobre, 2011; Kiyonaga and Egner, 2013; Postle, 2006; Zimmermann et al., 2016).’

In the Introduction, we changed ‘Recent models assume that this task-relevant information is stored…’ to: ‘It is further assumed that WM arises through the coordinated recruitment, via attention, of brain areas in a broad network (Constantinidis and Procyk, 2004; Postle, 2006; Ranganath and D'Esposito, 2005) and that the task-relevant information is stored…’.

As stated above, working memory load, or selective attention, during the delay was higher in task 4 than in task 5. Our finding that the M100 peak amplitudes evoked by the S1s in these two tasks were very similar suggests that the levels of general attention during S1 presentation were very similar in the two tasks and that the difference in working memory load or selective attention emerged after S1. To emphasize this point, we changed the sentence in the Results – ‘this suggests that levels of attention in tasks 4 and 5 were similar because the amplitude …’ – to: ‘this suggests that levels of general attention in tasks 4 and 5 were similar during the presentation of S1 because the peak amplitude…’.

With respect to general anticipation of upcoming S2 during the delay between S1 and S2, we agree with the reviewers that it could be different between high- and low-WM-load trials in tasks 1, 2, and 6. However, we consider it part of the difference in the preparation for motor responses because timing and anticipation of S2 were essential for preparing a correct go response which was required within a limited time window after S2 (for details, see Section 4.1.1), but not for a nogo response. To make this clear to readers, we added ‘within a limited time window after S2’ in the second paragraph of Section 2.1.1 and added ‘within a limited time window’ in the third paragraph of Setion 2.1.1.

Section 2.3.2, fourth paragraph: Were these significant ratios increases or decreases? Also, here, in the trained monkey L who performed tasks 1 and 6 on sequences where S1 and S2 were separated by longer delays of 1100 ms, the final 500 ms showed even more units that were significant. This could mean that there was more buildup of anticipation for the upcoming S2 as the animal (neurons) had to wait longer for the S2 presentation.

Both, significant increases and decreases were observed. We performed additional statistical tests to compare the percentage of units with significant ratios during the final 500 ms of the 1100-ms delay with that during the final 500 ms of the 800-ms delay. We found no significant differences. Our results therefore do not support the reviewers’ proposition that ‘there was more buildup of anticipation for the upcoming S2 as the animal (neurons) had to wait longer for the S2 presentation’. To make the comparison clearer, we added the statistics and also made the following changes.

In Section 2.3.2, we changed the sentence ‘Significant ratios during the final 500 ms of the 1100-ms delay were observed […] would decay over time’ to: ‘Significant ratios during the final 500 ms of the 1100-ms delay (23.2% and 18.8% of 69 recordings in tasks 1 and 6) were observed slightly, but not significantly more frequently (p>0.05 for tasks 1 and 6, chi-square test, one-tailed) than during the final 500 ms of the 800-ms delay (17.7% and 15.3% of 124 recordings). This result is inconsistent with that expected if significant ratios are due to differences in late activity because such differences would be expected to decay over time.’

Section 2.3.1, third paragraph: Please provide the data.

As requested, we provided the data for the second method (i.e., between-task comparison) by adding figure source data ([Supplementary-material SD1-data]) and making the following changes in the manuscript.

In Section 2.3.1, we changed ‘Results of this method were similar to those of the first method and are therefore not reported in detail here’ to: ‘Results of this method ([Supplementary-material SD1-data]) were consistent with those of the first method.’

In Source data legends we added: ‘[Supplementary-material SD1-data]. Working-memory-related neuronal activity identified by between-task comparison. […] The total number of recording sites was 307 for spike activity in core fields of AC, 310 for the LFP in core fields of AC, and 238 for spike activity in vlPFC.’

The data for the third method (i.e., comparing activity between the delay and the baseline in the same trial) have been partially reported in the original manuscript (see Figure 4, Figure 5 and Figure 6). We therefore do not provide additional data.

To help answer the question of whether anticipation/timing and attention contribute to the sustained delay activity or whether there is a ramping (up or down) of activity in anticipation of a meaningful S2, the reviewers recommend that you compare in the monkey studies the sessions which used ISIs of 800 ms versus those of 1100 ms.

As explained above, we performed the recommended comparison and found no significant difference (for details, see our responses to the comment ‘Section 2.3.2, fourth paragraph: Were these significant ratios increases or decreases? had to wait longer for the S2 presentation’). Questions related to anticipation and attention have been answered above as well.

3) Section 2.1.2, the sentence starting "However, the difference in the right AC…". If this difference was found beyond the final 500 ms of the delay, it may be influenced by the response or reward of being told one is correct. The conclusion that this reflects differences in working memory load does not seem well justified.

The difference beyond the final 500 ms of the delay was referred to the difference from 500 to 1000 ms of the delay (the first green bar in Figure 2 in our original manuscript) rather than the difference after S2. We apologize that we did not make this point clear. To avoid confusion, we deleted: ‘which were also found beyond the final 500 ms of the delay’.

4) It is unclear why a given neuron (or set of neurons) should exhibit working-memory related activity only in task 1 or in task 6, unless that is related to the frequency preferences of the neurons. Unfortunately, no information is provided about that. The frequencies of the two tones used in the monkey experiments were 1 and 3 kHz. It is stated in the Discussion (end of section 3.3) that not all neurons responded to these frequencies, but it is essential to provide the best or characteristic frequencies of the neurons in the Results section. Without this, the suggestion in the last paragraph of section 3.3 that the same neurons represent and store information becomes much weaker.

To strengthen our statement, we added the requested information about the neurons’ best frequencies in our manuscript.

In Section 2.3.2, we added: ‘The occurrence of significant spike-rate ratios in only one of the two tasks was related to the units’ best frequencies (for details of the assessment of the best frequency, see Sections 4.2.1 and 4.2.2). Significant spike-rate ratios only in task 1 occurred more frequently in units with a best frequency of ∼3 kHz (2.67-3.37 kHz) than in other units (26.7% vs. 9.4%; p<0.05, chi-square test, one-tailed). Significant spike-rate ratios only in task 6 occurred more frequently in units with a best frequency of ∼1 kHz (0.89-1.12 kHz) than in other units (30.1% vs. 15.4%; p<0.05).’

In Section 4.2.1, we added: ‘In an additional passive block, pure tones with 40 different frequencies were presented at ∼60 dB SPL to the monkeys to assess each unit’s best frequency. […] These tones were presented before the monkeys performed the behavioral tasks, or before the passive-condition measurements on days in which the monkeys did not perform the tasks.’

In Section 4.2.2, we added: ‘For each unit, the best frequency was assessed by analyzing the responses to the 40 pure tones as described by Brosch et al. (1999).’

5) No statistics appear to have been applied to the MEG data to test differences between high and low working memory loads, or any task-related signal vs. baseline. In the Results section, it is stated that "sources in the left and right AC were significantly stronger in high than in low WM load trials", but no statistics are presented to back this up.

We had applied statistical tests for differences in activity between high- and low-WM-load trials. This was mentioned in Section 4.1.2 (in our original manuscript). We apologize that this was not clear. Our statement that ‘sources in the left and right AC were significantly stronger in high than in low WM load trials’ was also backed up by statistics which were presented at the end of the sentence (see our original manuscript). To make it clearer to readers that statistics were applied, we made the following changes.

We added statistics in the manuscript. We also added p-values in Figure 2 and Figure 3 for each 500-ms period of the delay and deleted the green bars.

In Figure 2 legend, we changed the sentence ‘The green bars represent the 500-ms periods of the delay during which […] differed significantly from 1’ to: ‘The numbers on the abscissae are the p-values of permutation tests for the ratios of the source strength in high- to that in low-WM-load trials or in go to that in nogo trials during the three 500-ms periods of the delay.’

In addition, we now also performed statistical tests for differences in activity between task-related signals (i.e., source strength during the M100 period, and during the periods from 500 to 1000 ms, 1000 to 1500 ms, and 1500 to 2000 ms after S1 onset) and baseline. All differences were significant at the significance level of 0.001. To make it clear to readers, we added these statistics in the manuscript.

6) It should be made clear what the authors mean by "AC" or auditory cortex. Are they referring to primary auditory cortex? If so, they should not use this term for the MEG data since one cannot unequivocally localize MEG signals to a definitive cortical location. A similar issue is related to definitely stating that the source closest to the motor cortex is actually motor cortex. The most likely regions that would exhibit preparatory activity are premotor areas. Why didn't the authors attempt to perform a better source location for these regions based on anatomical images in a manner that was done for auditory cortex? Even in the monkey recordings, some justification for the claim that the recordings were localized to "core area (mainly from A1)" is required.

We agree with the reviewers that MEG signals cannot be unequivocally localized to a definite cortical location due to the ambiguity of the MEG-inherent inverse problem. Thus, as stated in our original manuscript, we used a different approach by directly seeding regional sources at a defined location, that is, at the border of Heschl’s gyrus and planum temporale (Näätänen and Picton, 1987), individually for each subject according to the anatomical MR image of the subject’s brain. This approach guarantees the consistency of source locations between subjects for computing the grand average. In monkey study, the term ‘auditory cortex’ referred to core fields of auditory cortex, mainly primary auditory cortex (A1). To make this difference between the human MEG studies and the monkey study clear, we kept the term ‘auditory cortex’ for our MEG studies and changed the term ‘auditory cortex’ to ‘core fields of auditory cortex’ and ‘AC’ to ‘core fields of AC’ in some places of the manuscript for our monkey study.

We added the information about how we justified the location of A1 in monkeys by in Section 4.2.2: ‘The position of A1 was estimated using the spatial distribution of the units’ best frequencies and the penetration traces of the electrodes, i.e., electrodes passed through parietal cortex into the auditory cortex (Brosch et al., 2005; Kaas and Hackett, 2000).’

We agree with the reviewers that, in our MEG studies, the two central probe sources provide only a weak estimation of activity in motor or premotor cortex. However, the main purpose of the two central probe sources, just as of the two frontal and two occipital probe sources, was to model background activity (Scherg, 1990; Scherg and Ebersole, 1993). The use of these probe sources is crucial, because otherwise AC sources could reflect activity not only in AC but also, to a certain extent, outside AC. All probe sources were placed very close to the cortical surface such that their distances to the AC source of the same hemisphere were relatively large. We did so on purpose because signals of sources influence each other and this mutual influence decreases with increasing distance between the sources.

Nonetheless, we followed the reviewers’ suggestions and moved the two central probe sources from the cortical surface into the hand regions of the motor cortices. We observed that signals of AC and motor cortex sources clearly intermingled due to their relatively small distance. These motor cortex sources showed an increase of their magnitude ∼100 ms after the onset of the auditory stimuli with a peak amplitude of ≥10 nAm. This observation indicates a strong influence of the signals from the AC sources. Thus, in order to minimize the mutual influence between the AC and probe sources, we decided to keep the probe sources close to the cortical surface.

In the course of moving the central probe sources into motor cortex, we detected, for few subjects, an error in our original manuscript, which occurred when exporting data files from BESA software to Matlab for further analysis. We apologize for this error and have contacted BESA company to issue a warning to prevent future users from repeating such an error. Our point has been confirmed by BESA and has been in the company’s issue tracking list (issue #370). We now have corrected this error in our revised manuscript and provided new Figure 2 . Fortunately, the corrected results are very similar to those presented in our original manuscript and provide an even more clear-cut support for our conclusions. In particular, there are no longer significant differences in neural activity during the delay between go and nogo trials in task 3. This means that the differences in activity between high- and low-WM-load trials in tasks 1 and 2 cannot be accounted for by differences in preparatory activity. Thus, there is no need to further investigate activity in motor or premotor cortex. We therefore deleted the figure panels illustrating the behavior of the central probe sources (corresponding to Figure 2 in our original manuscript) but include them in this letter (see Figure 8). Note that, compared to the original manuscript, the major finding of a stronger activity in go than in nogo trials during the final 500 ms of the delay in the left central source also did not change. Also note that, the strengths of the central sources were much weaker than those in auditory cortex.

Author response image 1.**DOI:**
http://dx.doi.org/10.7554/eLife.15441.015

Accordingly, we changed the Results sections in human studies 1 and 2, and listed the detailed changes in the following.

In Section 2.1.2, we changed the paragraphs ‘We found neural activity in the human AC that was related to WM […] likely reflect differences in WM load’ to the following: ‘We found neural activity in the human AC that was related to WM. […] In this task, differences between go and nogo trials were not significant (p>0.05, permutation test), neither in the left AC (Figure 2) nor in the right AC (Figure 2).’

In Section 2.2.2, we changed ‘However, in both ACs, the sources were significantly stronger in task 4 than in task 5 during considerable portions of the delay, including the final 500 ms (green bars, p<0.05/3, permutation test)’ to: ‘In addition, in both ACs, the sources were stronger in task 4 than in task 5 during the entire delay, being significant during many portions of the delay (p<0.05 with Bonferroni correction, permutation test; for p values, see the numbers above the abscissae).’

In Section 4.1.2, we added: ‘These three pairs of sources were located close to the cortical surface such that their distances to the source in the AC of the same hemisphere were relatively large. We did so to minimize the mutual influence between the signals of these sources and of the sources in AC because the mutual influence decreases with increasing distance between the sources.’

In Section 4.1.2, we deleted: ‘The central ECDs were also used to estimate activity in motor cortex’.

In Section 4.1.2, we changed ‘Statistical analyses were performed by permutation tests of the waveforms […] for multiple comparisons’ to: ‘Statistical analyses were performed by permutation tests. The significance level was set to 0.05.’

7) The MEG data from task 5 is very perplexing given that persistent activity above baseline is found during a delay period when no working memory is required. Can the authors provide any insight into this finding?

In task 5, the persistent activity during the delay period above baseline might be related to mental processes such as preparation for behavioral responses, and expectation of upcoming stimuli and rewards. These mental processes could be different between delay and baseline. As we stated in our original manuscript, activity in AC is not only related to processing sounds but also related to the aforementioned mental processes. These processes could be associated with changes in AC activity that last for seconds (for an example review, see Brosch et al., 2011).

8) In several human fMRI studies (e.g. Curtis, J. Neuro, 2004), delay period activity has been linked to behavior. Have the authors made any attempt to relate activity to behavior in either of their human or monkey data?

In our monkey study, we indeed related neuronal activity during the delay to behavior by comparing neuronal responses in correct and in error trials and by comparing neuronal responses in trials when the monkeys performed the tasks and in trials in the passive condition. These results were presented in Figure 4, Figure 5 and Figure 6 in our original manuscript. In our human studies, subjects’ performance was very high (~90% in study 1; ~95% in study 2). Hence, the number of error trials was too low to achieve, by averaging, a reasonable signal-to-noise ratio for comparison.

9) It is suggested in the first paragraph of section 3.4 that because working memory related activity was observed in a similar proportion of multiunits and single units, such neurons are likely to be organized in clusters. This is an extremely weak argument. Conclusions about local clustering can only be made if neighboring recordings sites show similar effects, whereas those located further apart do not.

We agree with the reviewers that this argument is weak. Nonetheless, we would like to discuss this possibility. We have made the following changes in the manuscript.

In Section 3.4, first paragraph, we changed ‘It is likely’ to ‘It is conceivable’.

In Section 3.4, first paragraph, we changed ‘This follows from our finding that multiunits displayed WM-related activity as common as single units (30% vs. 22%)’ to ‘This speculation is consistent with our finding that WM-related activity in multiunits was as common as in single units (30% vs. 22%)’.

In Section 3.4, first paragraph, we changed ‘should’ to ‘would’.

*10) The comparison between auditory cortex and vlPFC is potentially interesting, particularly given the surprising implication that the effects observed may not originate in PFC. However, this is based on one monkey only and so these findings are arguably premature. Since simultaneous recordings were made from these regions, was there any evidence for functional connectivity between them? If not, how can the authors be sure that they were recording in appropriate regions for PFC to precede auditory cortical activity? At very least, a recording placement figure for vlPFC and AC would add value to the manuscript for comparison to data from other labs.*

We indeed made some simultaneous recordings in the vlPFC and core fields of AC and found a synchronization between the two areas when the monkey performed the tasks. The vlPFC has also been demonstrated to be reciprocally connected with auditory cortex in a study combining microelectrode recording with anatomical tract-tracing (Romanski et al., 1999). In Section 4.2.2, we added this information: ‘We also found that neuronal activity in the vlPFC could be synchronized with that in auditory cortex, following the observation of Romanski et al. (1999) that the vlPFC reciprocally connected with auditory cortex.’

To further justify that the recordings were in the auditory area of the vlPFC, we added in Section 4.2.2: ‘This follows the observation of Romanski and Goldman-Rakic (2002) that auditory neurons were tightly clustered in a small region of vlPFC outside of which no auditory neurons were found.’

We added a figure supplement (Figure 1—figure supplement 3) to demonstrate the recording placement in the vlPFC and added in Figure 1—figure supplement 3 legend: ‘Figure 1—figure supplement 3. Demonstration of the recording area in prefrontal cortex. The red patch indicates the recording area on a lateral view of the macaque left hemisphere. The gray rids are the stereotactic coordinates according to Szabo and Cowan (1984). LS, Lateral Sulcus; PS, Principal Sulcus; AS, Accurate Sulcus.’

As explained above, we have used established criteria to justify that we made the recordings from core fields of AC. Because core fields in monkeys locate upon the lower bank of the lateral sulcus and are hidden from view by the overlying frontal and parietal lobes, a plot showing the entry points of the electrodes on cortical surface would not be informative to demonstrate that the recordings were made in core fields of AC. We therefore did not present such a figure for AC.

*Is the analysis of differential latencies in vlPFC and AC based only the 500 msec preceding S2, or was the entire delay including the S1 offset analyzed?*

The analysis of differential latencies was based on the entire delay from the offset of S1 to the onset of S2. We added this information in Section 2.3.3, last paragraph: ‘relative to the offset of S1,’.

*No statistics are provided in the fourth paragraph of section 2.3.3. Even if this is based on permutation tests, as mentioned in the methods, the details should be provided here.*

We added statistics here by making the following changes in the manuscript.

In Section 2.3.3, fourth paragraph, we added: ‘(p<0.05, permutation test)’. Other statistics related to comparison between experimental conditions were added in figures (Figure 4, Figure 5 and Figure 6). We added in Figure 4 legend: ‘The asterisks indicate significant differences between the conditions (chi-square test, one-tailed, p<0.001).’